# Inhibition gates supralinear Ca$^{2+}$ signaling in Purkinje cell dendrites during practiced movements

**Michael A Gaffield[1], Matthew J M Rowan[1], Samantha B Amat[1], Hirokazu Hirai[2], Jason M Christie[1]\***

[1]Max Planck Florida Institute for Neuroscience, Jupiter, United States; [2]Gunma University Graduate School of Medicine, Maebashi, Japan

**Abstract** Motor learning involves neural circuit modifications in the cerebellar cortex, likely through re-weighting of parallel fiber inputs onto Purkinje cells (PCs). Climbing fibers instruct these synaptic modifications when they excite PCs in conjunction with parallel fiber activity, a pairing that enhances climbing fiber-evoked Ca$^{2+}$ signaling in PC dendrites. In vivo, climbing fibers spike continuously, including during movements when parallel fibers are simultaneously conveying sensorimotor information to PCs. Whether parallel fiber activity enhances climbing fiber Ca$^{2+}$ signaling during motor behaviors is unknown. In mice, we found that inhibitory molecular layer interneurons (MLIs), activated by parallel fibers during practiced movements, suppressed parallel fiber enhancement of climbing fiber Ca$^{2+}$ signaling in PCs. Similar results were obtained in acute slices for brief parallel fiber stimuli. Interestingly, more prolonged parallel fiber excitation revealed latent supralinear Ca$^{2+}$ signaling. Therefore, the balance of parallel fiber and MLI input onto PCs regulates concomitant climbing fiber Ca$^{2+}$ signaling.

DOI: https://doi.org/10.7554/eLife.36246.001

## Introduction

Neural circuits that support motor learning must respond to, and adjust for, stimuli relevant for encoding adaptation. With equal importance, these circuits must also prevent network alterations during stimuli unrelated or unnecessary for behavioral modification. In the cerebellar cortex, synaptic re-weighting of parallel fiber input onto Purkinje cells (PCs) is thought to provide the basis for many types of motor learning (*Ito and Kano, 1982*; *Ito et al., 1982*; *Kano et al., 1992*; *Hansel and Linden, 2000*). Climbing fibers instruct synaptic alterations by evoking dendritic Ca$^{2+}$ spikes in PCs (*Kano et al., 1992*; *Hansel and Linden, 2000*; *Coesmans et al., 2004*) in response to adaptive stimuli (*Simpson and Alley, 1974*; *Gilbert and Thach, 1977*; *Medina and Lisberger, 2008*). However, climbing fiber activity also occurs regularly at 1–2 Hz (*Mukamel et al., 2009*; *Ozden et al., 2012*; *De Gruijl et al., 2014*). Therefore, PCs must distinguish which climbing fiber signals are relevant for inducing adaptation, as well as prevent or gate inappropriate circuit alterations by climbing fibers when plasticity is unwarranted or apt to produce an incorrect alteration of behavior (*Kimpo et al., 2014*). The integrative features of PC dendrites offer solutions as to how PCs accomplish this task (*Najafi and Medina, 2013*).

    When activated in conjunction with parallel fibers in ex vivo preparations, climbing fiber-triggered Ca$^{2+}$ signals in PC dendrites are enhanced in a supralinear manner (*Wang et al., 2000*; *Otsu et al., 2014*; *Piochon et al., 2016*). Supralinear signaling achieves a threshold level of intracellular Ca$^{2+}$ elevation that triggers long-term depression (LTD) at parallel fiber-PC synapses (*Finch et al., 2012*). Whether climbing fiber Ca$^{2+}$ signals in PC dendrites are augmented by preceding parallel fiber activity in vivo is unclear. Certainly, PCs receive sensorimotor information transmitted by granule cells

**\*For correspondence:**
jason.christie@mpfi.org

**Competing interests:** The authors declare that no competing interests exist.

during the execution of movements (*Ozden et al., 2012*; *Wilms and Häusser, 2015*; *Chen et al., 2017*). Furthermore, enhanced PC $Ca^{2+}$ signals are observed in response to externally produced sensory stimuli (*Najafi et al., 2014a*; *Najafi et al., 2014b*). However, if self-generated parallel fiber activity is sufficient to enhance ongoing climbing fiber $Ca^{2+}$ signals, then PCs would continuously undergo plasticity despite conditions where learning provides no benefit to motor outcomes. To counteract direct parallel fiber excitation of PCs, parallel fibers also excite molecular layer interneurons (MLIs) driving feed-forward inhibition that can attenuate parallel fiber excitatory postsynaptic potentials (EPSPs) (*Brunel et al., 2004*; *Mittmann et al., 2005*). MLIs can also directly reduce climbing fiber-evoked responses (*Callaway et al., 1995*; *Kitamura and Häusser, 2011*) and impair LTD at parallel fiber-PC synapses (*Ekerot and Kano, 1985*). Consequently, the balance of excitatory and inhibitory input onto PCs may determine the level of dendritic $Ca^{2+}$ signaling and, ultimately, the extent of climbing fiber-dependent learning.

We examined climbing fiber-evoked $Ca^{2+}$ signals in PC dendrites in vivo to determine how these responses are regulated by local circuit activity in the cerebellar cortex of awake behaving mice. During the performance of practiced movements, we found that parallel fiber and climbing fiber co-activity failed to produce supralinear $Ca^{2+}$ signals. However, disinhibiting the molecular layer through chemogenetic suppression of MLI activity enhanced the amplitude of climbing fiber-evoked $Ca^{2+}$ signals in PCs, specifically during movements when both parallel fibers and MLIs were activated. Quantitative ex-vivo measurements in PC dendrites confirmed that MLI-mediated feed-forward inhibition limits the ability of parallel fiber excitation to produce supralinearity. Bi-directional optogenetic actuation of MLI activity during parallel fiber stimuli altered associative parallel fiber-climbing fiber $Ca^{2+}$ signaling, dependent on the level of MLI output. Our results show that climbing fiber $Ca^{2+}$ signals in PCs are regulated by the counterbalance of MLI-mediated inhibition with parallel fiber-evoked EPSPs.

## Results

### Climbing fiber $Ca^{2+}$ signals in PCs are unresponsive to behavior-induced parallel fiber activity

We used two-photon laser scanning microscopy (2pLSM) to measure climbing fiber-evoked $Ca^{2+}$ activity in PC dendrites of lobule Crus II in head-fixed mice conditioned to lick for water from a port when cued by an audible tone (*Figure 1A*). After practice, expert mice reliably produced orofacial movements on command (*Figure 1B*; see also *Gaffield and Christie, 2017*). Climbing fiber-evoked $Ca^{2+}$ events were continuously apparent in PCs transduced with the genetically encoded $Ca^{2+}$ indicator GCaMP6f using an AAV vector under control of a truncated version of the PC-specific *Pcp2* promoter (*Chen et al., 2013*; *Nitta et al., 2017*). During the initiation of licking, the frequency of $Ca^{2+}$ events increased nearly five-fold, indicating engagement of the lateral cerebellum during water consumption (*Gaffield et al., 2016*). This behavior also elicited the activity of granule cell parallel fibers, as determined in separate $Ca^{2+}$ imaging experiments from GCaMP6f-transduced neurons in the same area of Crus II (*Figure 1—figure supplement 1A–D*). As with MLIs in this region (*Gaffield and Christie, 2017*), average parallel fiber $Ca^{2+}$ activity closely tracked lick rate (*Figure 1—figure supplement 1E*), indicating encoding of licking-related kinematics in their population as observed in other cerebellar lobules (*Jelitai et al., 2016*; *Chen et al., 2017*; *Giovannucci et al., 2017*; *Knogler et al., 2017*).

To examine for an influence of parallel fibers on climbing fiber-evoked $Ca^{2+}$ signaling in PCs, we collected well-isolated individual dendritic $Ca^{2+}$ events from PCs (*Figure 1C*). Events were categorized whether they occurred during water consumption when granule cells were active or in the absence of licking movements when granule cells were relatively inactive. Isolated events comprised the majority of all identified $Ca^{2+}$ responses in PC dendrites, whether or not the animal was actively engaged in water consumption (*Figure 1—figure supplement 2A*). Surprisingly, we found no difference in the average amplitudes of isolated $Ca^{2+}$ events across PCs between these two behavioral states ($\Delta F/F = 0.223 \pm 0.008$ and $0.223 \pm 0.011$, licking and non-licking, respectively; p=0.99; N = 11 mice; paired Student's t-test; *Figure 1D*). Similar results were also obtained for all other PC $Ca^{2+}$ events; this included overlapping responses occurring closely in time (*Figure 1—figure supplement 2B and C*). The lack of behavioral-dependent differences in PC $Ca^{2+}$ event amplitudes was unlikely

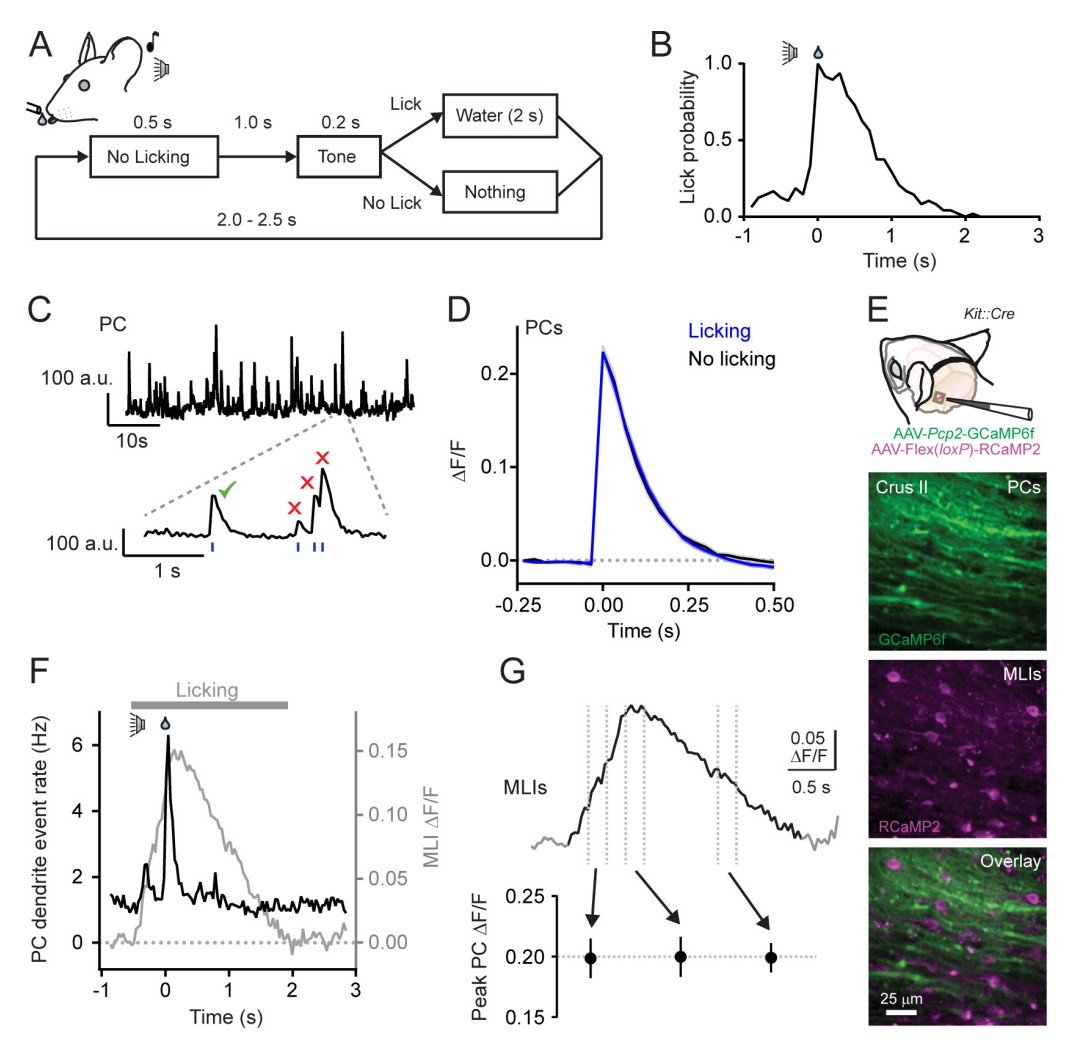

**Figure 1.** Climbing fiber-evoked Ca$^{2+}$ signals in PCs during behavior. (**A**) Head-fixed mice were trained to lick water from a port, cued by an audible tone, using the procedure shown. (**B**) Across-trial distribution of lick probability, aligned to the delivery of water, for a trained mouse. (**C**) Continuous record of Ca$^{2+}$ activity in a PC dendrite. Expanded area shows algorithmically identified climbing fiber-evoked events (blue tick marks). Isolated Ca$^{2+}$ events, indicated by the checkmark, were collected for analysis. (**D**) Average of climbing fiber-evoked Ca$^{2+}$ events in PC dendrites occurring during water consumption (blue) or in the absence of licking (black). Measurements were obtained from 11 to 51 PCs in each of 11 mice; 211 cells total. (**E**) Genetic targeting of PCs and MLI using AAVs with GCaMP6f under control of the *Pcp2* promoter and Cre-dependent RCaMP2 in *Kit::Cre* mice. In vivo images are from an infected area of Crus II. (**F**) The average frequency of climbing fiber-evoked Ca$^{2+}$ events in PC dendrites (11 to 19 PCs in each of 5 mice; 82 cells total) plotted against the response in MLIs, acquired simultaneously in a subset of recordings (3 mice). (**G**) Trial-averaged measurement of MLI activity during cued licking. The peak amplitudes of Ca$^{2+}$ events in PCs, plotted below, that correspond to three different phases of MLI activation during the task (4 to 19 PCs in each of 6 mice; 79 cells total; p=0.99, ANOVA test).

DOI: https://doi.org/10.7554/eLife.36246.002

The following source data and figure supplements are available for figure 1:

**Source data 1.** Source data for panels D and G.
DOI: https://doi.org/10.7554/eLife.36246.008

**Figure supplement 1.** Ca$^{2+}$ activity measurements in parallel fibers.
DOI: https://doi.org/10.7554/eLife.36246.003

**Figure supplement 1—source data 1.** Source data for panels B and E.
DOI: https://doi.org/10.7554/eLife.36246.004

**Figure supplement 2.** Analysis of non-isolated PC dendritic Ca$^{2+}$ events.

*Figure 1 continued*

DOI: https://doi.org/10.7554/eLife.36246.005

**Figure supplement 2—source data 1.** Source data for panels A-C.

DOI: https://doi.org/10.7554/eLife.36246.006

**Figure supplement 3.** PC Ca$^{2+}$ event amplitudes do not co-vary with the level of MLI activity.

DOI: https://doi.org/10.7554/eLife.36246.007

attributable to GCaMP6f biophysics. While this Ca$^{2+}$ indicator is nonlinear (*Chen et al., 2013*), it has been shown to be sensitive enough to report slight alteration of PC Ca$^{2+}$ signals evoked by unexpected sensory stimuli (*Najafi et al., 2014b*). Thus, climbing fiber-evoked Ca$^{2+}$ signaling in PC dendrites appeared resistant to licking-related co-activity of granule cells. This also suggests that PC dendritic Ca$^{2+}$ responses are not always subject to enhancement in the context of behavior (*Najafi et al., 2014a*; *Najafi et al., 2014b*) and that the influence of parallel fiber excitation on the integrated dendritic response to climbing fiber input may be subject to regulation.

In addition to PCs, granule cells also excite MLIs. Therefore, it was not surprising that, using in vivo Ca$^{2+}$ imaging, we found that the licking-induced activation of MLIs showed a close correspondence to that of parallel fibers, measured simultaneously, during licking (*Figure 1—figure supplement 1B*). This suggests a potential regulatory counterbalance of MLI-mediated feed-forward inhibition onto PCs in superimposition with their direct excitation by granule cells. If so, activation of MLIs during licking could alter conjunctive parallel fiber-climbing fiber Ca$^{2+}$ signaling in PC dendrites. To examine the relationship between MLIs and climbing fiber-evoked responses in PCs, we used a dual-color imaging approach to simultaneously measure Ca$^{2+}$ activity in both cell types. By performing experiments in *Kit::Cre* mice, a driver line with high specificity for MLIs (*Amat et al., 2017*; these mice may have a low abundance of Golgi cell targeting), we could transduce these cells with an AAV containing Cre-dependent RCaMP2 (*Inoue et al., 2015*), a red genetically encoded Ca$^{2+}$ indicator spectrally separable from green GCaMP6f expressed in PCs (*Figure 1E*). During bouts of cued licking, climbing fibers evoke Ca$^{2+}$ signals in PCs at the same time that the ensemble of surrounding MLIs was activated by parallel fibers (*Figure 1F*). However, the amplitude of climbing fiber-evoked Ca$^{2+}$ events in PCs did not co-vary with the activity level of MLIs (*Figure 1G* and *Figure 1—figure supplement 3*) that increased and decreased in proportion to adjustments in lick rate during water consumption (*Gaffield and Christie, 2017*). In conclusion, the amplitude of climbing fiber-evoked Ca$^{2+}$ responses in PC dendrites appeared unaffected by movement when both parallel fibers and MLIs were active.

## Disinhibition enhances climbing fiber-evoked Ca$^{2+}$ signals in PCs during parallel fiber activity

To determine whether MLI-mediated inhibition influences climbing fiber-evoked Ca$^{2+}$ signals in PC dendrites, we chemogenetically suppressed the activity of MLIs and imaged Ca$^{2+}$ responses in PCs during cued bouts of licking (*Figure 2A*). For this approach, we injected AAV containing Cre-dependent hM4d (*Armbruster et al., 2007*) into left Crus II of *Kit::Cre* mice to transduce MLIs and co-expressed GCaMP6f selectively in PCs using AAVs under control of the truncated *Pcp2* promoter (*Figure 2B*). Intraperitoneal injection of clozapine-N-oxide (CNO), the cognate agonist of hM4d, led to an increase in the amplitude of isolated climbing fiber-evoked dendritic Ca$^{2+}$ events in PCs relative to control measurements obtained in separate sessions (*Figure 2C and F*). Similar results were also apparent for all other non-isolated Ca$^{2+}$ events as well (*Figure 2—figure supplement 1A and B*). Importantly, the effect of this chemogenetic manipulation was conditional, occurring only when MLIs were active. Whereas climbing fiber-evoked Ca$^{2+}$ events were enhanced during the consumption of water, events occurring in the absence of licking movements were unaffected by disinhibition of the molecular layer (*Figure 2D and F*). In a separate set of experiments performed on a cohort of *Kit::Cre* mice lacking hM4d expression in MLIs, CNO administration had no effect on Ca$^{2+}$ event amplitudes during water consumption (*Figure 2E and F*). This result rules out the possibility that an off-target influence of the drug accounted for the alteration of dendritic Ca$^{2+}$ signaling.

A complementary analysis showed that molecular layer disinhibition by chemogenetics increased trial-averaged Ca$^{2+}$ activity in PCs (*Figure 2—figure supplement 2A and B*). This unbiased measure

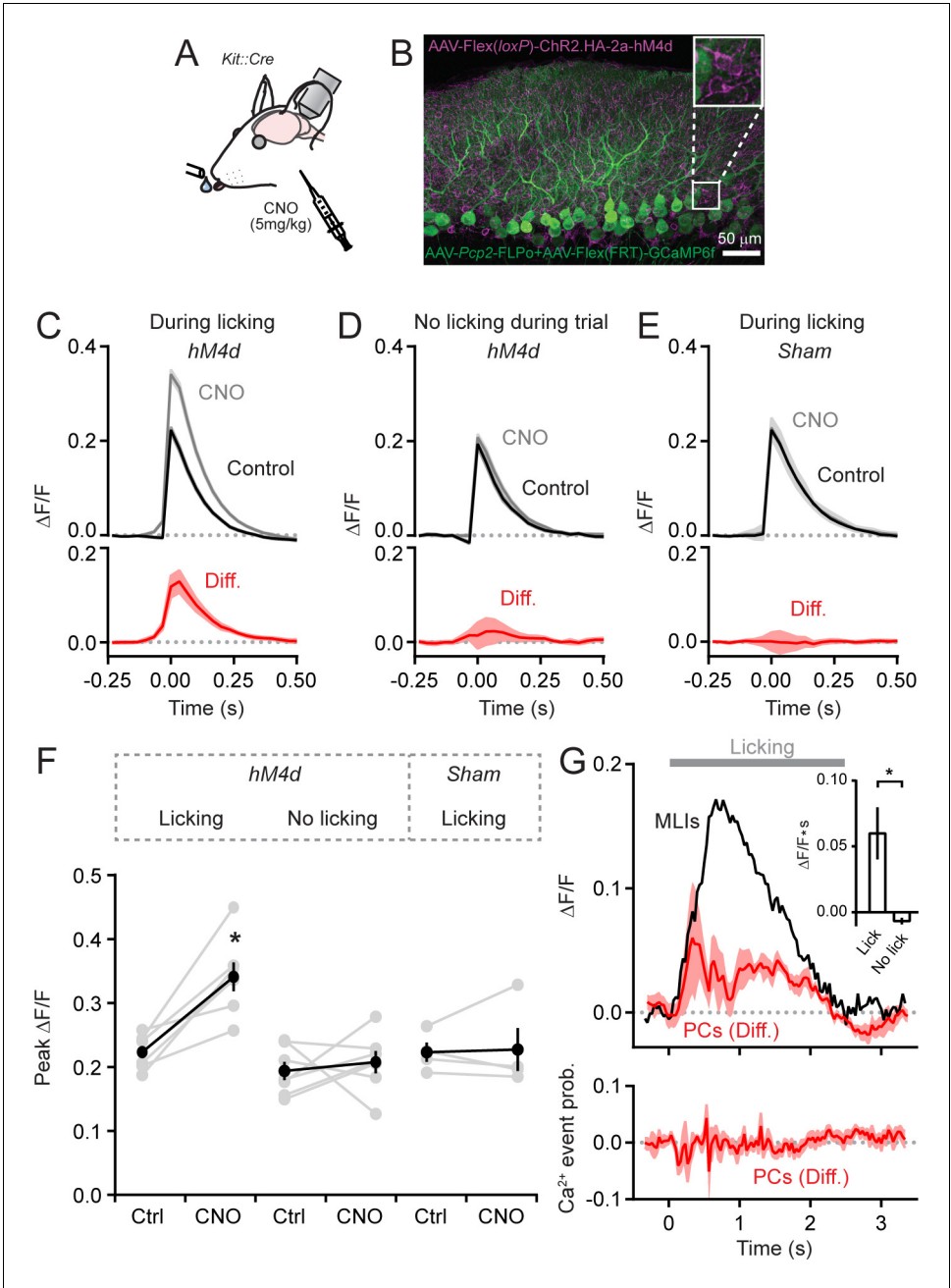

**Figure 2.** MLIs suppress climbing fiber-evoked dendritic $Ca^{2+}$ signals in PCs during licking. (**A**) Following an audible cue, head-fixed mice licked from a port during in vivo imaging with 2pLSM. In some sessions, CNO was administered by intraperitoneal injection prior to the start of the task. (**B**) Image of fixed tissue showing PCs transduced with GCaMP6f and MLIs expressing HA-tagged hM4d (inset is a magnified view of labeled MLI somata). For this set of experiments, PCs were transduced with AAV containing the recombinase FLPo under control of the PC-specific *Pcp2* promoter in combination with an FLPo-dependent AAV containing GCaMP6f to generate high-level expression of the $Ca^{2+}$ indicator. (**C–E**) Average dendritic $Ca^{2+}$ events in PCs recorded in control or in sessions with CNO (11 to 35 PCs each from 7 mice, 100 and 156 cells total; control and CNO, respectively). Isolated $Ca^{2+}$ events were sorted based on whether or not they occurred in correspondence with licking. A separate group of sham mice lacked expression of hM4d were also tested (10 to 51 PCs each from 4 mice, 111 and 100 cells total; control and CNO, respectively). Difference signals are shown below in red. (**F**) Summary plot of the effect of chemogenetic MLI activity suppression on the amplitude of isolated dendritic $Ca^{2+}$ events in PCs. Black, group average (mean ± SEM); gray, individual mice (N = 7). Asterisk indicates a significant difference from all other conditions (p<0.01; ANOVA with Tukey's post-hoc multiple comparison test; all other

*Figure 2 continued on next page*

*Figure 2 continued*

comparisons were insignificant). (G) Top plot, the difference in trial-averaged PC dendritic $Ca^{2+}$ activity between sessions in control and in CNO. Note the correspondence of the difference signal in PCs and the licking-evoked activation of MLIs measured simultaneously in a subset of experiments. Inset: the mean integrated $Ca^{2+}$ activity during water consumption (lick) and in the absence of licking (no lick). Asterisk indicates a significant difference (p=0.025; paired Student's t-test). Bottom plot, the probability of climbing fiber-evoked $Ca^{2+}$ events in PCs was unchanged by molecular layer disinhibition, relative to that in control (11 to 35 PCs each from 6 mice, 145 and 97 cells total).

DOI: https://doi.org/10.7554/eLife.36246.009

The following video, source data, and figure supplements are available for figure 2:

**Source data 1.** Source data for panels C-E and G.
DOI: https://doi.org/10.7554/eLife.36246.016
**Figure supplement 1.** MLI-mediated suppression of non-isolated PC $Ca^{2+}$ events.
DOI: https://doi.org/10.7554/eLife.36246.010
**Figure supplement 1—source data 1.** Source data for panels A and B.
DOI: https://doi.org/10.7554/eLife.36246.011
**Figure supplement 2.** Chemogenetic suppression of MLIs increases trial-averaged $Ca^{2+}$ activity in PC dendrites.
DOI: https://doi.org/10.7554/eLife.36246.012
**Figure supplement 2—source data 1.** Source data for panels A and B.
DOI: https://doi.org/10.7554/eLife.36246.013
**Figure supplement 3.** Chemogenetic suppression of MLIs in Crus II does not affect licking rates.
DOI: https://doi.org/10.7554/eLife.36246.014
**Figure supplement 3—source data 1.** Source data for panels B and C.
DOI: https://doi.org/10.7554/eLife.36246.015
**Figure 2—video 1** Licking behavior in control and the disinhibited condition.
DOI: https://doi.org/10.7554/eLife.36246.017

reflects the integration of all dendritic $Ca^{2+}$ during licking bouts independent of any event detection. The enhancement of trial-averaged dendritic $Ca^{2+}$ activity was time-locked to the licking-evoked activation of MLIs, measured simultaneously in a subset of these experiments (*Figure 2G*). The effect of disinhibition on trial-averaged PC $Ca^{2+}$ activity is likely attributable to an increased amplitude of climbing fiber events because chemogenetic suppression of MLI activity did not affect the rate of $Ca^{2+}$ events in PCs (1.55 ± 0.19 Hz and 1.51 ± 0.47 Hz for control and CNO, respectively; p=0.51, paired Student's t-test; *Figure 2G*) and climbing fiber inputs are known to produce the majority of $Ca^{2+}$ elevation in dendrites in response to in vivo excitation (*Mukamel et al., 2009*; *Ozden et al., 2009*; *Najafi et al., 2014b*).

Our previous work showed that bilateral chemogenetic suppression of MLIs in Crus II slows the rate of licking, indicating an influence of MLIs on motor output (*Gaffield and Christie, 2017*). However, in these current experiments, we limited expression of hM4d to MLIs of left Crus II which, upon suppression by CNO administration, failed to significantly affect average licking dynamics (rate, pattern, rhythmicity, and time from cue presentation to lick initiation; *Figure 2—figure supplement 3A–C*, *Figure 2—video 1*). Although this result argues against the possibility that chemogenetic-induced behavioral perturbations accounted for alteration of climbing fiber-evoked responses in PCs, it may be that unquantified orofacial movements, such as lateral tongue displacement, were altered. We attempted to control for this more carefully by also examining PC $Ca^{2+}$ activity in a subset of hM4d-expressing mice that had closely matched licking rates before and during molecular layer disinhibition. Even in these animals, isolated $Ca^{2+}$ event amplitudes were increased with MLI activity suppression (*Figure 2—figure supplement 3D*). We conclude that alteration of climbing fiber-evoked $Ca^{2+}$ signaling in PCs with disinhibition resulted from the influence of MLIs on PC dendritic integration. In summary, these results show that MLI-mediated inhibition recruited during motor behavior (*Jelitai et al., 2016*; *Astorga et al., 2017*; *Gaffield and Christie, 2017*) suppresses climbing fiber-evoked $Ca^{2+}$ signaling in PCs dendrites. Hence, the integration of information transfer from the inferior olive to the cerebellar cortex, encoded in the $Ca^{2+}$ activity of PCs will depend, in part, on the output of MLIs.

## Disinhibition affects climbing fiber-evoked Ca²⁺ signaling throughout PC dendrites

We determined the extent to which MLI-mediated inhibition affects $Ca^{2+}$ signaling across the activated ensemble of PCs. For this analysis, we used movement-related difference images of averaged PC dendritic activity obtained from sessions performed in the disinhibited condition normalized to that obtained in control sessions for each mouse (*Figure 3A*). More than 70% of identified PC dendrites exceeded a threshold level of altered activity indicative of enhancement during licking movement with MLI activity suppression (*Figure 3B*). Therefore, MLIs influence $Ca^{2+}$ signaling in the majority of activated PCs likely owing to the widespread and coherent activation of these interneurons in lobule Crus II during orofacial motor behavior (*Astorga et al., 2017*; *Gaffield and Christie, 2017*).

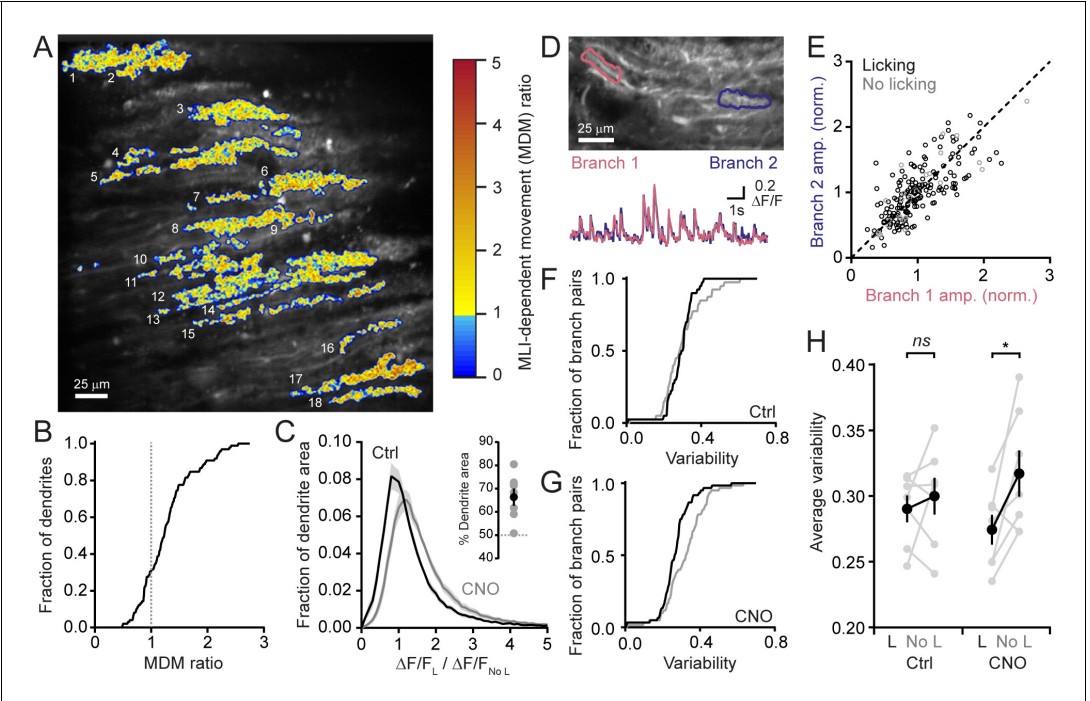

**Figure 3.** MLIs broadly influence climbing fiber-evoked $Ca^{2+}$ signaling in PCs. (**A**) The across-session change in trial-averaged PC $Ca^{2+}$ activity, colored-coded based on the extent to which chemogenetic disinhibition affected responses during movement (MDM ratio; see Materials and methods). Each algorithmically identified PC dendrite is numbered. (**B**) Cumulative probability histograms of the effect of disinhibition on trial averaged $Ca^{2+}$ activity in identified PCs (2 to 15 PCs each from seven mice, 50 cells total). Dotted line demarcates a threshold level of change indicative of enhancement with disinhibition (MDM ratio >1). (**C**) Histogram of average $Ca^{2+}$ activity measurements, determined for each dendritic pixel across the PC population, during licking (L) divided by that observed in the absence of licking (No L), in both control sessions (black) as well as after administration of CNO (gray). For all mice (N = 7), the distributions were significantly different (p<0.0001, Kolmogorov-Smirnov test). The inset shows the summary of ROC analysis on these distributions, obtained for each mouse, where area under the curve was used to calculate the percentage of pixels that showed an effect with chemogenetic disinhibition (gray, individual mice; black, the mean ± SEM). (**D**) In the image, two segments of a dendrite from distinct branches of the same PC are outlined. Measurements of $Ca^{2+}$ activity in these segments are shown superimposed in the traces below. (**E**) Comparison of the amplitudes of simultaneous, inter-branch $Ca^{2+}$ events for many climbing fiber-evoked responses for the two branches shown in example in panel D. Events were sorted based on whether they occurred during water consumption (black) or in the absence of licking (gray). Unity is marked by the dashed line. (**F,G**) Cumulative probability of the inter-branch variance of climbing fiber $Ca^{2+}$ event amplitudes (see Materials and methods). Events were sorted depending on their correspondence with water consumption (black) or the absence of licking (gray). Distributions were not different in control sessions (N = 74 pairs; range: 4 to 20 pairs each from seven mice; p=0.57, Kolmogorov-Smirnov test) but were in the disinhibited condition with CNO (N = 114 pairs; range 12 to 28 pairs each from seven mice; p=0.0013, Kolmogorov-Smirnov test). (**H**) The effect of molecular layer disinhibition on average inter-branch variability of $Ca^{2+}$ event amplitudes in PCs. In control sessions, the variance was similar whether or not events occurred during licking movements (p=0.50; paired Student's t-test). In contrast, a modest, but significant drop in variability occurred during movement with disinhibition (p=0.029; paired Student's t-test). Black, group average (mean ± SEM); gray, individual mice.

DOI: https://doi.org/10.7554/eLife.36246.018

A similar analysis was used to measure for subcellular effects of MLI-mediated inhibition on dendritic $Ca^{2+}$ signals in PCs. First, we determined the average change in $Ca^{2+}$ activity due to movements during cued water consumption relative to baseline measurements in the absence of licking for each pixel in all identified dendrites. This was calculated for the same PCs in both control sessions as well as during chemogenetic activity suppression of MLIs. There was a clear shift to larger values in the disinhibited condition (*Figure 3C*). Next, from these distributions, we used receiver operating characteristic (ROC) curves to estimate that ~66% of the fractional area of all individual dendrites showed enhanced $Ca^{2+}$ signaling with MLI activity suppression (*Figure 3C*, inset). This indicates that enhancement occurred throughout individual dendrites suggesting that MLIs produce a widespread influence on PC $Ca^{2+}$ signaling.

PCs receive input from many MLIs that make distributed synaptic contacts onto their dendrites (*Palay and Chan-Palay, 1974*; *Kim et al., 2014*). By producing localized inhibitory effects on climbing fiber-evoked dendritic spiking, MLIs can contribute to increased variability of climbing fiber-evoked $Ca^{2+}$ signals in the arbors of individual PCs (*Callaway et al., 1995*; *Kitamura and Häusser, 2011*). Comparisons of climbing fiber $Ca^{2+}$ activity in two distinct branches of the same PC dendrite showed that, although evoked responses occurred reliably at both locations, their amplitudes could differ slightly (*Figure 3D*). This held true whether or not the animal was consuming water (*Figure 3E,F and H*). However, after MLI activity suppression by intraperitoneal injection of CNO, there was a modest decrease in the inter-branch variability during licking movements (*Figure 3G*) that was statistically different (*Figure 3H*). This points to a potential non-uniform effect of MLI-mediated inhibition on the PC response to climbing fiber excitation during task engagement. In conclusion, MLIs broadly influence climbing fiber-evoked $Ca^{2+}$ signaling in PC dendrites during a practiced motor behavior.

## Enhanced PC $Ca^{2+}$ signaling is not attributable to an alteration in climbing fiber activity

It remains possible that the enhancement of $Ca^{2+}$ signaling in PCs following molecular layer disinhibition reflects an increase in presynaptic climbing fiber activity which is translated into a larger postsynaptic response. This is because burst firing of olivary projection neurons (*Crill, 1970*), conveyed to the cerebellar cortex by climbing fibers, promotes increased PC dendritic spiking and larger amplitude $Ca^{2+}$ signals (*Mathy et al., 2009*; *Kitamura and Häusser, 2011*). To assess this possibility, we directly measured the activity of climbing fibers using $Ca^{2+}$ imaging. We injected AAV containing GCaMP6f under control of the CaMKIIα promoter into the inferior olive transducing excitatory projection neurons (*Mathews et al., 2012*) as evidenced by transgene expression in climbing fiber axons in the cerebellar cortex (*Figure 4A*). In recordings from awake animals, we used automated routines (*Hyvärinen, 1999*) to group like-responding pixels from images obtained in the molecular layer. The resulting segments comprised individual, sagittally-aligned climbing fibers (*Figure 4B*).

$Ca^{2+}$ events occurred regularly in climbing fibers with mean rates comparable to the frequency of dendritic events in PCs, measured in separate animals (*Figure 4B and C*), and within the range of previously published PC event rates (*Mukamel et al., 2009*; *Ozden et al., 2012*; *De Gruijl et al., 2014*). Therefore, in awake mice, climbing fiber activity reliably drives dendritic spiking in their postsynaptic targets. The amplitudes of $Ca^{2+}$ events in individual climbing fibers showed considerable variation during ongoing activity (*Figure 4B*). Sorted events, collected across mice, had a non-normal distribution (p<0.0001, Shapiro-Wilk Test; N = 3579 events; 30 climbing fibers; 7 mice). Instead, amplitude distributions skewed towards larger values suggestive of a multimodal composition (*Figure 4D*). Our interpretation of these observations is that $Ca^{2+}$ events in climbing fibers are generated by discrete, high-frequency (100–400 Hz) bursts of firing and that the variance in their amplitudes reflects heterogeneity in the number of action potentials contained in the burst (*Mathy et al., 2009*).

The frequency of $Ca^{2+}$ events in climbing fibers increased at licking onset (*Figure 4E*). This indicates that licking initiation is signaled in their activity and generates the corresponding uptick in dendritic $Ca^{2+}$ events in PCs during the same period of the task (see *Figure 1F*). To evaluate whether burst firing in climbing fibers encodes licking-related information, we compared $Ca^{2+}$ events evoked during water consumption to those occurring in the absence of licking (*Figure 4F*). The lack of significant differences between the amplitudes of mean events in these conditions (p=0.89; paired Student's t-test) indicates that the spike content of presynaptic bursts in climbing fibers varies

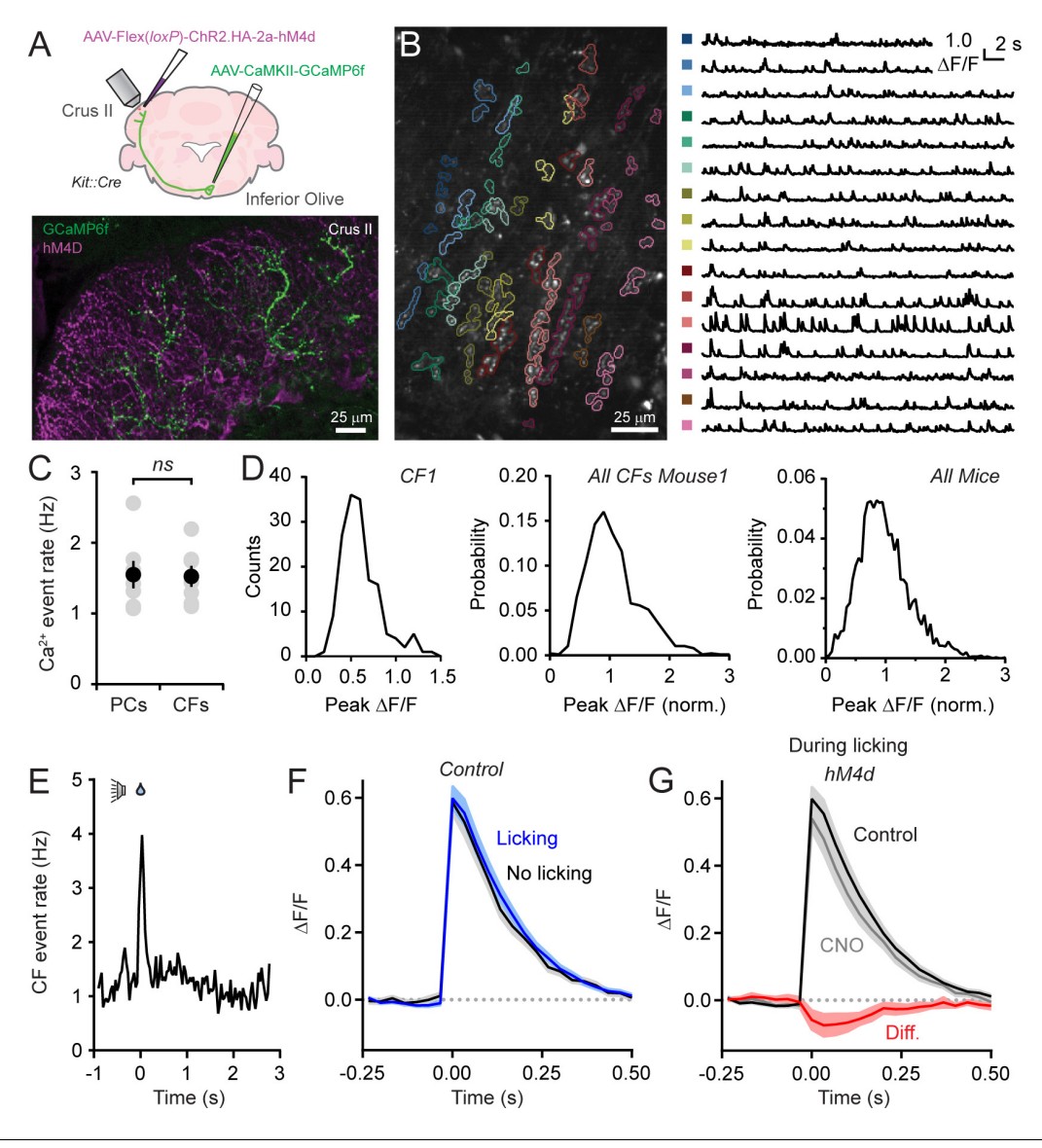

**Figure 4.** Disinhibition does not affect presynaptic climbing fiber activity. (**A**) AAVs containing genetically encoded activity reporters and effectors were injected in the inferior olive and lobule Crus II of *Kit::Cre* mice, respectively. Image from fixed tissue showing GCaMP6f expression in climbing fibers and HA-tagged hM4d in MLIs. (**B**) In the image, individual climbing fibers were identified using automated segmentation routines. Traces show activity measurements from color-coded climbing fibers. (**C**) $Ca^{2+}$ event rates in PC dendrites and climbing fibers, measured in separate cohorts of mice (11 to 19 PCs and 2 to 6 climbing fibers each from 7 mice, 100 and 29 total, respectively). Black circles, mean ± SEM; gray circles, measurements from individual mice (p=0.92, Student's t-test). (**D**) Distribution of $Ca^{2+}$ event amplitudes for an individual climbing fiber, all climbing fibers in a single mouse (N = 6), and for all mice (2 to 12 climbing fibers each from 7 mice, 36 fibers total). Data were normalized to facilitate comparisons across climbing fibers. (**E**) The frequency of $Ca^{2+}$ events in climbing fibers during cued licking (average of 3 mice). (**F**) Average of isolated $Ca^{2+}$ events in climbing fibers collected either during the consumption of water (blue) or in the absence of licking (black). Measurements obtained from 4 to 12 climbing fibers each from 5 mice, 38 fibers total. (**G**) $Ca^{2+}$ events recorded in climbing fibers both in control and during sessions with chemogenetic MLI activity suppression. Events were collected only during periods of water consumption (4 to 9 climbing fibers each from 5 mice, 26 fibers total). The difference signal is shown in red.
DOI: https://doi.org/10.7554/eLife.36246.019

The following source data is available for figure 4:

**Source data 1.** Source data for panel C.
DOI: https://doi.org/10.7554/eLife.36246.020

independent of practiced movements. Because these measurements were performed in *Kit::Cre* mice expressing hM4d in MLIs, we disinhibited the molecular layer by intraperitoneal injection of CNO (*Figure 4A*) and assessed for differences in climbing fiber $Ca^{2+}$ events. Chemogenetic suppression of MLI activity had no effect on $Ca^{2+}$ event rates (change of 15.5 ± 16.2% from control, p=0.47, paired Student's t-test) nor did it affect the amplitude of events evoked during licking when MLIs are normally activated (p=0.66, paired Student's t-test; *Figure 4G*). This result rules out that enhancement of PC dendritic $Ca^{2+}$ signaling with molecular layer disinhibition is due to a change in presynaptic climbing fiber activity. Furthermore, it argues against the possibility that unresolved, closely-spaced $Ca^{2+}$ events in postsynaptic PCs, erroneously categorized as individual responses, accounted for the amplitude change with MLI activity suppression. Otherwise, this would have also been reflected as a corresponding increase in $Ca^{2+}$ event amplitudes in presynaptic climbing fibers.

## MLIs suppress supralinear climbing fiber-evoked $Ca^{2+}$ signaling in PC dendrites

The amplitude of climbing fiber-evoked $Ca^{2+}$ signals in PC dendrites can be enhanced if climbing fibers are stimulated in conjunction with preceding parallel fiber activity (*Wang et al., 2000*). This supralinearity may reflect a change in dendritic excitability that facilitates the propagation of climbing fiber-evoked $Ca^{2+}$ spikes into spiny branchlets yielding additional $Ca^{2+}$ entry (*Otsu et al., 2014*; but see *Wang et al., 2000*). Parallel fibers also activate MLIs, driving rapid feed-forward inhibition that attenuates parallel fiber excitation of PCs (*Brunel et al., 2004*; *Mittmann et al., 2005*). We reasoned that, by reducing dendritic excitability, feed-forward inhibition could diminish the ability of parallel fibers to enhance subsequent climbing fiber-evoked $Ca^{2+}$ responses and thus provide a mechanism to explain in vivo gating of non-linear dendritic $Ca^{2+}$ signaling in PCs during licking movements.

To quantitatively assess this possibility, we measured $Ca^{2+}$ activity in individual PC dendritic branches using 2pLSM in acute cerebellar slices from mature *Kit::Cre* mice. Experiments were performed in the absence of synaptic blockers while rapidly and reversibly suppressing MLI firing with high temporal precision using the anion-fluxing channelrhodopsin *Gt*ACR2 (*Govorunova et al., 2015*), transduced in MLIs by Cre-dependent AAV (*Figure 5A* and *Figure 5—figure supplement 1A*). In whole-cell PC recordings, electrical stimulation of parallel fibers adjacent to the PC dendrite evoked depolarizing post-synaptic potentials (PSPs) that were prolonged when MLI activity was optogenetically prevented with a pulse of blue light coincident with the parallel fiber stimulus (*Figure 5B and C* and *Figure 5—figure supplement 1B and C*). The effect of optogenetic MLI activity suppression on the parallel fiber-evoked PSP integral was indistinguishable from that produced following pharmacological block of GABA$_A$ receptor-mediated transmission, indicating that *Gt*ACR2 completely prevented feed-forward inhibition (*Figure 5—figure supplement 1D*).

Electrical stimulation of climbing fibers evoked complex spikes in the PC soma (*Figure 5B*) and accompanying $Ca^{2+}$ transients in their spiny dendrites, resolved by including the $Ca^{2+}$ indicator Fluo-5F in the patch pipette (*Figure 5D*). Pairing a brief parallel fiber tetanus in conjunction with the climbing fiber stimulus (50 ms interval) produced dendritic $Ca^{2+}$ responses that were no different than the expected summed combination of the individual parallel fiber and climbing fiber transients, computed from separately evoked responses on alternate trials (100.5 ± 2.1% of sum; N = 31, p=0.78; paired Student's t-test; *Figure 5D*). Changing the timing of the preceding parallel fiber tetanus, relative to the conjunctive climbing fiber stimulus, failed to uncover supralinear $Ca^{2+}$ signaling (range: 25–100 ms; *Figure 5—figure supplement 2A*). However, when we optogenetically suppressed feed-forward inhibition during the parallel fiber tetanus in interleaved trials, the amplitude of the climbing fiber-evoked $Ca^{2+}$ transient was greater than the estimated summed response (109.9 ± 2.4% of sum; N = 31; p=0.0003; paired Student's t-test; *Figure 5D*). Thus, supralinear $Ca^{2+}$ signaling in PCs could be uncovered with molecular layer disinhibition.

On average, enhancement of climbing fiber-evoked $Ca^{2+}$ signaling by parallel fibers in the absence of MLI feed-forward inhibition was greatest for short intervals and decayed to non-significance for long intervals (*Figure 5—figure supplement 2B*). Thus, supralinear $Ca^{2+}$ signaling was dependent on the timing of parallel fiber and climbing fiber activity (*Wang et al., 2000*; *Brenowitz and Regehr, 2005*). Although, even at short intervals, the effect of disinhibition on supralinear $Ca^{2+}$ signaling varied to some extent across dendrites (*Figure 5E*). This suggests that the temporal sensitivity of parallel fiber-climbing fiber interactions may be set not only by the functional

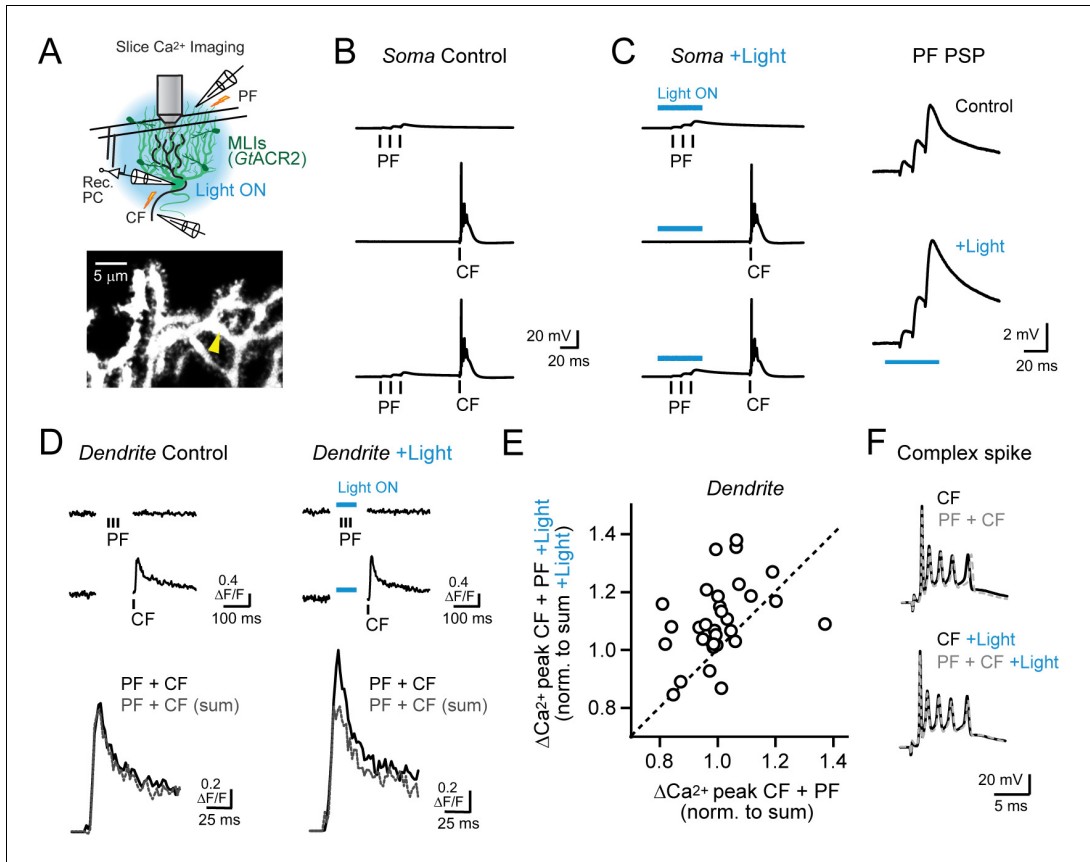

**Figure 5.** Feed-forward inhibition attenuates supralinear Ca$^{2+}$ signaling in PC dendrites. (A) In acute slices, parallel fibers were stimulated in conjunction with climbing fibers during whole-cell patch recording from PCs in lobule Crus II. Ca$^{2+}$ imaging was performed using 2pLSM in PC spiny dendrites as shown in the fluorescence image. (B) Evoked responses in a PC either to the parallel fiber tetanus (3 pulses at 100 Hz), the climbing fiber stimulus, or their conjunctive pairing (50 ms interval). (C) In alternate trials, MLIs expressing *Gt*ACR2 were photoinhibited using wide-field illumination with blue light (λ = 461 nm; 40 ms; 6.6 mW/mm$^2$), coincident with the parallel fiber tetanus. Average parallel fiber-evoked PSPs in control and during optogenetic suppression of feed-forward inhibition are enlarged on the right. (D) Left, the average climbing fiber-evoked Ca$^{2+}$ signal in a PC dendrite (location demarcated by the yellow arrowhead in the morphological image in panel A) produced following conjunctive stimulation with parallel fibers. The estimated summed response, shown in gray, for parallel fiber and climbing fiber transients evoked in isolation on separate trials (Ca$^{2+}$ activity traces shown above). Right, climbing fiber-evoked Ca$^{2+}$ signals from the same dendritic location but with feed-forward inhibition suppressed by optogenetics during the parallel fiber tetanus. (E) The change in amplitude of climbing fiber-evoked Ca$^{2+}$ signals with conjunctive stimulation of parallel fibers, measured in the same PC dendrite, in trials either with or without optogenetic suppression of MLI-mediated feed-forward inhibition. Data are normalized to the estimated, summed response of parallel fibers and climbing fibers for each condition. Each point is a measurement from a different dendritic branch (3 to 10 sites for each of 5 PCs, 31 sites total) with unity demarcated by the dashed line. (F) Somatic complex spikes evoked by the climbing fiber stimulus, both in isolation as well as in conjunction with parallel fiber activation. Responses in the same PC in control and with MLIs photo-inhibited during the parallel fiber tetanus.

DOI: https://doi.org/10.7554/eLife.36246.021

The following source data and figure supplements are available for figure 5:

**Figure supplement 1.** Optogenetic elimination of MLI-mediated feed-forward inhibition.
DOI: https://doi.org/10.7554/eLife.36246.022

**Figure supplement 2.** Temporal dynamics of conjunctive parallel fiber-climbing fiber Ca$^{2+}$ signaling in PC dendrites.
DOI: https://doi.org/10.7554/eLife.36246.023

**Figure supplement 2—source data 1.** Source data for panel B.
DOI: https://doi.org/10.7554/eLife.36246.024

region of the cerebellar cortex (*Suvrathan et al., 2016*) but also locally at the level of individual synapses. The number of spikelets in the somatic complex spike burst was unaffected by suppression of feed-forward inhibition (4.2 ± 0.2 and 4.5 ± 0.3 spikelets; control and with disinhibition, respectively; N = 6; p=0.17; Student's t-test; *Figure 5F*) pointing to the compartmentalized influence of MLI activity on climbing fiber signaling in PC dendrites (*Callaway et al., 1995*). Together, these results indicate that, with feed-forward inhibition intact, brief parallel fiber activation failed to enhance climbing fiber-evoked Ca$^{2+}$ signals likely due to the attenuating influence of MLI-mediated inhibition on the parallel fiber EPSP. However molecular layer disinhibition revealed the latent ability of parallel fibers to enhance climbing fiber-evoked dendritic Ca$^{2+}$ signaling, similar to our in vivo findings.

## Activity-dependent recovery of supralinear PC dendritic Ca$^{2+}$ signaling by parallel fibers

Granule cells encode sensorimotor information conveyed through the mossy fiber pathway, with their activation level dependent on self-produced and external stimuli. This includes enhanced firing in response to the multimodal integration of many mossy fiber input streams (*Ishikawa et al., 2015*; *Giovannucci et al., 2017*). To further examine if associative climbing fiber-evoked Ca$^{2+}$ signaling in PC dendrites is sensitive to the level of parallel fiber activity, we increased the number of stimuli in the parallel fiber tetanus (*Figure 6A and B*). With a more prolonged tetanus, conjunctive climbing fiber-evoked Ca$^{2+}$ signals were of greater amplitude than the estimated summed responses of parallel fibers and climbing fibers alone (*Figure 6B and C*; also observed in a matched subset of observations at the same dendritic site: 3 PF stimuli, ΔF/F 101.1 ± 2.5% of expected linear sum; 9 PF stimuli, ΔF/F 114.7 ± 3.6%; p=0.005; N = 14; paired Student's t-test). Such supralinear Ca$^{2+}$ signaling was apparent in the majority of PC dendrites examined, in contrast to that observed in separate PC recordings using a tetanus with fewer parallel fiber stimuli (*Figure 6C and D*). Thus, with a sufficient level of parallel fiber activation, the resulting direct excitation of PCs can overwhelm feed-forward inhibition to recover supralinear Ca$^{2+}$ signaling. This indicates that the balance of dendritic excitation and inhibition through the mossy fiber pathway is a critical determinate of conjunctive parallel fiber-climbing fiber Ca$^{2+}$ signaling in PCs.

We tested this possibility more rigorously in acute cerebellar slices prepared from *Kit::Cre* mice infected with a Cre-dependent AAV vector containing the excitatory red-shifted channelrhodopsin bReaChES (*Rajasethupathy et al., 2015*). This allowed us to optogenetically increase the inhibitory output of transduced MLIs during the prolonged parallel fiber tetanus (*Figure 6A*). Supralinear Ca$^{2+}$ signaling in PC dendrites evoked by prolonged parallel fiber and climbing fiber conjunctive stimuli (ΔF/F 121.5 ± 5.3% of the expected linear sum of both inputs) was abolished in alternate trials when MLIs were optogenetically activated coincident with the parallel fiber tetanus (98.0 ± 3.5% of summed response; p=0.004; N = 17; paired Student's t-test; *Figure 6E and F*; *Figure 6—figure supplement 1A and B*). In a subset of experiments, MLI activity was systemically varied during the same recordings using different photostimulus intensities in alternating trials. Supralinear climbing fiber Ca$^{2+}$ signals in PC dendrites were reduced, dependent on the activity level of MLIs during the parallel fiber tetanus (*Figure 6G*). This supports the hypothesis that the relative activity of parallel fibers and MLIs not only determines whether coincident parallel fiber and climbing fiber activity produces supralinear Ca$^{2+}$ signaling in PC dendrites, but also the magnitude of the enhancement as well.

## Discussion

Using a combination of Ca$^{2+}$ imaging and genetically encoded effectors of activity, we find that inhibitory MLIs exert a profound regulatory influence on climbing fiber-evoked Ca$^{2+}$ signaling in PC dendrites. In awake mice engaged in a routine motor task that activates parallel fibers, MLIs enforce normalization of dendritic climbing fiber-evoked Ca$^{2+}$ signals matching those occurring spontaneously during quiescence when parallel fibers are inactive. In ex vivo recordings, short bursts of parallel fiber stimuli fail to evoke supralinear climbing fiber Ca$^{2+}$ signals in PCs dendrites due to MLI-mediated feed-forward inhibition that attenuates parallel fiber EPSPs. Thus, during the performance of practiced movements, recruitment of inhibition from MLIs gates climbing fiber-evoked Ca$^{2+}$ signaling that might otherwise induce plasticity and, therefore, is well positioned to constrain learning in the absence of motor performance errors.

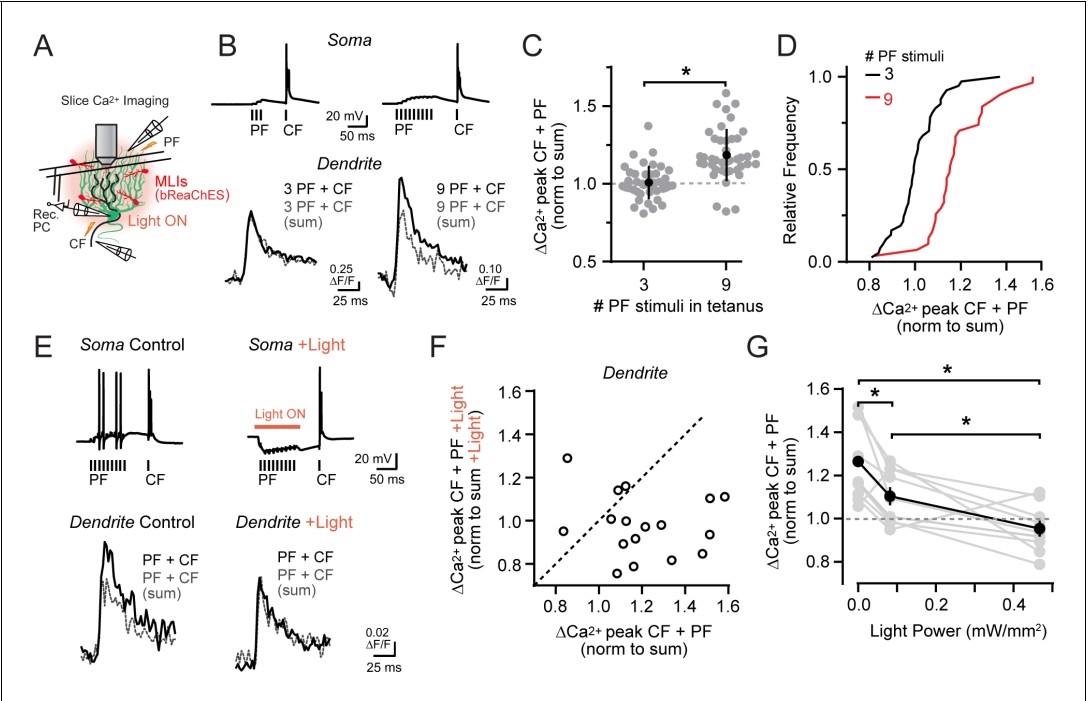

**Figure 6.** Activity-dependent recovery of supralinear climbing fiber Ca²⁺ signaling is sensitive to MLI inhibition. (**A**) Acute slice recording configuration. In a subset of experiments, bReaChES was expressed in MLIs by Cre-dependent AAV in *Kit::Cre* mice. (**B**) Comparison of average climbing fiber-evoked Ca²⁺ signals in two different PCs with the conjunctive parallel fiber tetanus including either 3 or 9 stimuli (100 Hz). The summed response of the parallel fiber and climbing fiber transients, evoked in isolation on separate trials, is shown in gray. (**C**) Across-cell comparison shows that increasing the number of stimuli in the parallel fiber tetanus results in a supralinear enhancement of climbing fiber Ca²⁺ signals in PC spiny dendrites. Individual dendritic recording sites are indicated in gray (N = 41 and 42 dendritic sites from 6 and 4 cells; 3 and 9 stimuli, respectively) with mean data in black (±SEM; p<0.0001; Student's t-test). (**D**) In the cumulative probability histogram, supralinear climbing fiber Ca²⁺ signaling was observed in a majority of PC dendrites when stimulated with a longer lasting parallel fiber tetanus. (**E**) Climbing fiber-evoked Ca²⁺ signals at the same PC dendritic site. In interleaved trials, optogenetic activation of MLIs (λ = 596 nm; 40 ms; 0.93 mW/mm²) occurred during the parallel fiber tetanus. (**F**) Relationship between the change in amplitude of climbing fiber-evoked Ca²⁺ signals with conjunctive activity of parallel fibers, in trials either with or without optogenetic activation of MLIs during the parallel fiber tetanus (3 to 7 dendritic sites from 4 cells, 17 sites total). Data are normalized to the estimated, summed response of the parallel fiber and climbing fiber transients for each condition recorded at the same PC dendritic location. Dashed line is unity. (**G**) The effect of varying optogenetic MLI-mediated inhibition on the ability of parallel fibers to produce supralinear climbing fiber Ca²⁺ signals in PC dendrites. Mean data (black symbols ± SEM) are from matched comparisons at the same dendritic recording site (4 to 5 sites from 2 cells, 9 dendrites total; p<0.05; Repeated measures 1-way ANOVA with Tukey's post hoc multiple comparison test). Gray symbols are individual measurements.

DOI: https://doi.org/10.7554/eLife.36246.025

The following figure supplement is available for figure 6:

**Figure supplement 1.** Influence of optogentic MLI activation on PC dendritic and somatic excitation.

DOI: https://doi.org/10.7554/eLife.36246.026

## Encoding of climbing fiber-evoked Ca²⁺ signals in PCs

Parallel fiber excitation drives simple spiking in PCs while convergent inhibition from MLIs influences the frequency and regularity of these responses (*Häusser and Clark, 1997*; *Mittmann et al., 2005*; *Dizon and Khodakhah, 2011*). In addition, climbing fiber-mediated activation of dendritic voltage-gated Ca²⁺ channels produces local, regenerative spikes in PC dendrites that are subject to amplification by preceding parallel fiber activity (*Wang et al., 2000*; *Schmolesky et al., 2002*; *Otsu et al., 2014*). Our observations indicate that, in behaving mice, climbing fiber-evoked Ca²⁺ signaling in PC dendrites is resistant to enhancement by co-active parallel fibers due to inhibition from MLIs. Therefore, in addition to influencing the pattern and timing of PC simple spiking during movements (*Jelitai et al., 2016*; *Chen et al., 2017*), MLIs also participate in encoding the Ca²⁺ response to climbing fiber excitation by locally gating the dynamics of non-linear dendritic signaling. We propose that by performing these non-mutually exclusive operations, MLIs are well-placed to influence motor control (*Heiney et al., 2014*) as well as motor learning (*Jörntell et al., 2010*).

Parallel fiber-evoked EPSPs inactivate subthreshold, voltage-gated K$^+$ currents ($I_{SA}$) in PC dendrites that can facilitate the initiation and propagation of subsequent climbing fiber-evoked Ca$^{2+}$ spikes into spiny branchlets, thereby enhancing intracellular Ca$^{2+}$ entry (*Otsu et al., 2014*). In our slice recordings, we observed that MLI feed-forward inhibition diminished the ability of parallel fibers to enhance climbing fiber-evoked Ca$^{2+}$ responses. This effect is likely attributable to the marked reduction in parallel fiber-triggered excitatory potentials by feed-forward inhibition (*Brunel et al., 2004*; *Mittmann et al., 2005*). Attenuated PSPs are expected to be less efficacious in generating depolarization-induced K$^+$ channel inactivation. However, in our acute slice experiments, the sensitivity of climbing fiber-evoked Ca$^{2+}$ signaling to enhancement by parallel fibers could be restored by longer lasting parallel fiber stimuli. Although, this effect could be offset, dependent on the level of MLI activity. Thus, gating of non-linear dendritic signaling in PCs is not necessarily an 'all-or-none' phenomenon as classically described for direct climbing fiber excitation (*Eccles et al., 1966*). Rather, the average amplitude of Ca$^{2+}$ signals in PC dendrites elicited by climbing fibers will depend on the net balance of preceding parallel fiber excitation and MLI-mediated inhibition, with the accumulation and recovery of $I_{SA}$ inactivation evolving with the summating PSPs. Our analysis did not distinguish whether, at the single trial level, the balance of PF-mediated excitation and inhibition influences the probability or the amplitude of a supralinear Ca$^{2+}$ event in PC dendrites, though both are apt to affect plasticity induction.

In our in vivo recordings, the amplitude of climbing fiber-evoked Ca$^{2+}$ events in PC dendrites did not co-vary with the level of MLI activation during water consumption. This implies that a homeostasis is achieved between the excitatory activity of parallel fibers and inhibition from MLIs to prevent supralinear climbing fiber Ca$^{2+}$ signaling. Even under conditions where MLI activity decreased during motor behavior (e.g., as the lick rate decreased), excitation and inhibition appeared in balance, as determined by the high correspondence of parallel fiber and MLI activity. This likely limited any enhancement of PC Ca$^{2+}$ events during task performance. Whether feed-forward inhibition can continuously counteract parallel fiber excitation across a range of behavioral conditions is yet to be determined.

Apart from regulating supralinear enhancement of Ca$^{2+}$ signals by parallel fibers, MLIs can also reduce the amplitude of climbing fiber-evoked Ca$^{2+}$ transients through direct electrogenic suppression of PC dendritic spiking (*Callaway et al., 1995*; *Kitamura and Häusser, 2011*). We cannot rule out the possibility that this mechanism also affects climbing fiber-evoked Ca$^{2+}$ signaling during practiced movements. However, direct suppression of dendritic spiking by inhibition would result in smaller amplitude Ca$^{2+}$ events during movement when MLIs are active compared to spontaneous events occurring in quiescence. We did not observe such an effect in our in vivo measurements. Therefore, we conclude that the predominant role of MLIs on climbing fiber-mediated Ca$^{2+}$ signaling during practiced motor behavior is through their inhibitory influence on non-linear dendritic operations in PCs.

## Climbing fiber Ca$^{2+}$ signaling and plasticity

Ca$^{2+}$ signals produced in the PC dendrite by climbing fiber excitation have been subject to intensive investigation because intracellular Ca$^{2+}$ elevation is a biochemical trigger for inducing synaptic plasticity (*Finch et al., 2012*; *Lamont and Weber, 2012*), and synaptic plasticity has been implicated as a neural correlate of motor learning and memory (*Marr, 1969*; *Albus, 1971*; *Ito and Kano, 1982*). When activated alone, climbing fiber-evoked Ca$^{2+}$ signals are insufficient to induce synaptic reweighting of parallel fiber-PC synapses. However, amplification of climbing fiber signals by preceding parallel fiber activity can achieve a threshold level of Ca$^{2+}$ elevation necessary to generate short- and long-term plasticity at parallel fiber inputs (*Wang et al., 2000*; *Brenowitz and Regehr, 2005*; *Tanaka et al., 2007*). Therefore, by preventing parallel fiber enhancement of Ca$^{2+}$ signaling, MLI-mediated inhibition may suppress the induction of climbing fiber-mediated plasticity despite the interaction of these two inputs.

In the absence of MLI constraint during behavior, parallel fiber enhancement of climbing fiber Ca$^{2+}$ signaling would occur continuously during self-generated movement. This would result in the continuous induction of climbing fiber-mediated plasticity at co-active parallel fiber inputs. Unconstrained plasticity in the disinhibited cortex may force parallel fiber-PC synapses into a saturated state where motor learning can no longer occur (*Nguyen-Vu et al., 2017*). Interestingly, genetic deletion of GABA$_A$Rs from PCs impedes consolidation of motor learning (*Wulff et al., 2009*), a

result that may be attributable to the disinhibition of PCs and indiscriminate plasticity produced by unregulated, supralinear climbing fiber Ca²⁺ signaling.

Inhibitory control of plasticity by modulation of dendritic excitability may be a ubiquitous function of GABA-releasing interneurons in the brain. Interneurons in the cortex, hippocampus, amygdala, and striatum are known to dynamically regulate dendritic processing in their postsynaptic targets, electrogenically gating synaptic interactions that lead to alterations in circuit function (*Paulsen and Moser, 1998*; *Letzkus et al., 2015*). In this report, we extend inhibitory regulation to non-linear Ca²⁺ signaling in PCs. Although inhibition can elicit branch-and synapse-specific control of Ca²⁺ signaling (*Callaway et al., 1995*; *Chiu et al., 2013*), our results indicate that during behavior, MLIs exert a widespread suppression of supralinear climbing fiber Ca²⁺ signaling throughout the PC dendrite, perhaps owing to the highly coherent activation of their ensemble during orofacial movements (*Astorga et al., 2017*; *Gaffield and Christie, 2017*). Even in experiments where molecular layer disinhibition reduced inter-branch variance of climbing fiber-evoked Ca²⁺ responses, the effect was small. This suggests that heterogeneity of MLI-mediated inhibition on Ca²⁺ signaling within PC dendritic arbors is limited during practiced movement. Therefore, we envision that inhibition gates plasticity on a cell-wide scale.

## Behavioral significance of MLI-gating of Ca²⁺ signaling in PCs

A novel aspect of our findings is that Ca²⁺ responses in PC dendrites are not always augmented by the context of active motor behavior. However, the lack of supralinear Ca²⁺ signaling during movement in our study does not preclude the possibility that such responses could occur under different behavioral conditions. In our experiments, mice elicited well-rehearsed motor responses that, through prior practice, resulted in highly stereotyped licks trial after trial indicative of few performance errors (*Gaffield and Christie, 2017*). During these trials, as well as during un-cued licking with free access to water, climbing fiber-evoked Ca²⁺ activity increases in PCs at the onset of water consumption (*Gaffield et al., 2016*). Importantly, we have not yet determined the role these signals play in the cued licking task, nor whether they (or the cerebellum in general) are required for the behavior. That the amplitudes of these evoked Ca²⁺ events are indistinguishable from spontaneous activity suggests that they may not be useful for learning. As the mice are performing a skilled behavior, these movement-evoked Ca²⁺ events could help with motor memory retention (*Medina et al., 2002*). During more mistake-prone behaviors, where mice must learn to carefully articulate their movements using the benefit of sensorimotor associations, it may be that parallel fibers contribute to climbing fiber-evoked Ca²⁺ signaling to reach a threshold necessary for inducing plasticity and modification of motor output. In this way, we do not discount the possibility that the extent of climbing fiber-mediated excitation of PCs, which may vary dependent on the severity of motor errors, plays a role in determining the magnitude of learning outcomes (*Yang and Lisberger, 2014*).

This hypothesis is consistent with work showing that external sensory cues that can guide associative learning (e.g., classical eyeblink conditioning), evoke graded climbing fiber-evoked Ca²⁺ signals in PC dendrites (*Najafi et al., 2014a*; *Najafi et al., 2014b*). Enhancement of these signals may be driven, in part, from conjunctive activity of parallel fibers (*Giovannucci et al., 2017*). In this scenario, enhanced climbing fiber-evoked Ca²⁺ signaling in PCs during learning may arise from alterations in network activity within the cerebellar cortex. Spike bursting of granule cells could shift the balance of parallel fiber excitation and feed-forward inhibition through short-term plasticity (*Häusser and Clark, 1997*; *Grangeray-Vilmint et al., 2018*), perhaps unlocking the ability of parallel fibers to generate non-linear Ca²⁺ signaling in PCs when activated by multimodal streams of sensorimotor information, as occurs during associative learning (*Chadderton et al., 2004*; *Ishikawa et al., 2015*; *Giovannucci et al., 2017*). In addition, MLIs inhibit one another through their structured GABAergic interconnections (*Kim et al., 2014*; *Rieubland et al., 2014*). Increased inhibition between MLIs in response to salient learning events would allow parallel fibers to generate supralinear Ca²⁺ signals in PCs during climbing fiber excitation. Alternatively, sensorimotor signals useful for producing learning could be conveyed to PCs through a separate population of granule cells whose parallel fibers bypasses MLIs (*Ekerot and Jörntell, 2001*). Future work will help clarify if and how MLI regulation of climbing fiber Ca²⁺ signaling in PCs is altered during learning.

An important caveat to our study is that we did not specifically monitor granule cell activity in the disinhibited state. Chemogenetic suppression of some Golgi cells, which may be expected

considering that *Kit::Cre* mice are not pristinely selective for MLIs (**Amat et al., 2017**), might produce enhanced excitability in a subpopulation of granule cells. We cannot rule out a scenario where such enhancement of PF signaling onto PCs - not apparent in licking behavior nor in the responsiveness of MLIs whose activity we monitored - directly contributes to increased PC dendritic $Ca^{2+}$ activity. Notably, the absence of a disinhibitory effect on the behavior-induced MLI population response following CNO administration may reflect the inability of our GCaMP imaging approach to discern the contribution of spike-driven $Ca^{2+}$ entry from activity mediated by synaptic sources (e.g., $Ca^{2+}$ permeable AMPA and NMDA receptors); the latter is expected to be relatively less susceptible to alteration by chemogenetic MLI activity suppression. However, strong hM4d-mediated suppression of MLI neurotransmission, apart from spiking and synaptic activation (**Stachniak et al., 2014**; **Amat et al., 2017**), combined with evidence from our ex vivo experiments showing a direct inhibitory influence of MLIs on PF-evoked enhancement of PC dendritic $Ca^{2+}$ signaling, support our conclusion that MLI activity gates PC responses to climbing fiber excitation in vivo.

In summary, our results emphasize the importance of the PC dendrite as a central locus for encoding the integrated response to climbing fiber input (**Najafi and Medina, 2013**) and, hence, determining the physiological consequence of olivary signaling in the cerebellar system. Climbing fibers fire in response to motor errors. Climbing fibers also fire during normal movements, activity that may be important for coordinating motor timing (**Lang et al., 2017**). Context-specific MLI activity might allow climbing fibers to functionally multiplex. Complex spikes in the PC soma could be transmitted to downstream premotor targets and influence motor control apart from $Ca^{2+}$ signaling in the PC dendrite where plasticity is induced. By gating supralinear climbing fiber-evoked $Ca^{2+}$ signaling, molecular layer inhibition may prevent unwarranted or unnecessary adaptation during accurately performed movements.

# Materials and methods

**Key resources table**

| Reagent type or resource | Designation | Source or reference | Additional information |
|---|---|---|---|
| strain, (Mus Musculus) | *Kit::Cre* | **Amat et al., 2017** | on C57Bl/6 background |
| transfected construct | AAV1-*Pcp2*.4-GCaMP6f | University of North Carolina | custom |
| transfected construct | AAV1-*Pcp2*.4-FLPo | University of North Carolina | custom |
| transfected construct | AAV1-CAG-Flex(FRT) rev-RCaMP2 | University of North Carolina | custom |
| transfected construct | AAV1-CAG-Flex(*loxP*) rev-RCaMP2 | University of Pennsylvania | custom |
| transfected construct | AAV1-CAG-Flex(FRT) rev-GCaMP6f | University of North Carolina | custom |
| transfected construct | AAV1-CAG-Flex(*loxP*) rev-ChR2.HA-2a-hM4d | ViGene | custom |
| transfected construct | AAV1-Syn-GCaMP6f | University of Pennsylvania | AV-1-PV2822 |
| transfected construct | AAV1-CaMKIIα-GCaMP6f | University of Pennsylvania | AV-1-PV2822 |
| transfected construct | AAV1-EF1α-Flex(*loxP*) rev-GtACR2.eYFP | ViGene | custom |
| transfected construct | AAV5-EF1α-Flex(*loxP*) rev-bReachES-TS-YFP | University of North Carolina | shelf |
| antibody | anti HA | Abcam | #ab9110 |
| software, algorithm | Prism | GraphPad | Statistical analysis |
| software, algorithm | Matlab | Mathworks | Image analysis |
| software, algorithm | ImageJ | NIH | Image analysis |

*Continued on next page*

*Continued*

| Reagent type or resource | Designation | Source or reference | Additional information |
|---|---|---|---|
| software, algorithm | bControl | Carlos Brody, Princeton | Behavior control |
| software, algorithm | ScanImage | Vidrio Technologies | Microscope control |

## Animals

Animal procedures were conducted using protocol 15–205 approved by the Institutional Animal Care and Use Committee (IACUC) at the Max Planck Florida Institute for Neuroscience. Heterozygous adult *Kit::Cre* mice (*Amat et al., 2017*) of both genders were used for all experiments (in vivo and ex vivo:>10 and>7 weeks of age, respectively).

## Surgical procedures

As described previously (*Gaffield et al., 2016*), cranial windows for in vivo imaging in the cerebellum were prepared from mice under isoflurane (1.5–2.0%). Warmth was provided by a heating pad using biofeedback to maintain a stable core body temperature (37°C). Non-responsiveness to intermittent toe pinches confirmed the surgical plane of anesthesia. For this procedure, the skull was exposed through surgical excision of the scalp (subcutaneous injection of lidocaine/bupivacaine provided local anesthesia). A custom-engineered stainless steel head post was then attached onto the dried, exposed bone, centered on the midline of the cranium, using Metabond (Parkell, Edgewood, NY). A small craniotomy (~2 mm square) was cut over the left lateral cerebellum using a scalpel without disturbing the underlying dura mater. The opening was covered with a small glass coverslip (CS-3R, Warner Instruments, Hamden, CT), and cemented in place with Metabond such that the window applied minimal pressure to the brain. Post-operative analgesia (buprenorphine; 0.35 mg/kg) was administered and the animal recovered under supervision until ambulatory.

Prior to placement of the window, adeno-associated viruses (AAVs) were pressure injected into the brain using beveled glass micropipettes. Viruses included: AAV1-Pcp2.4-GCaMP6f, AAV1-Pcp2.4-FLPo, AAV1-CAG-Flex(FRT)rev-RCaMP2, AAV1-CAG-Flex(*loxP*)rev-RCaMP2, AAV1-CAG-Flex(FRT)rev-GCaMP6f, AAV1-CAG-Flex(*loxP*)rev-ChR2.HA-2a-hM4d, AAV1-Syn-GCaMP6f (all custom prepared at the University of North Carolina Vector Core Facility, the University of Pennsylvania Vector Core Facility, or ViGene, Rockville, MD). For injections into lobule Crus II, the micropipette generally contained multiple viruses (100–150 nl) in order to avoid repeated penetrations of the same location. Three different injection depths (150–350 µm below the dura) were used to evenly transduce both PCs and MLIs. To transduce granule cells, AAV1-Syn-GCaMP6f was injected 250–350 mm below the dura (*Giovannucci et al., 2017*); this non-selective approach also resulted in expression of the $Ca^{2+}$ indicator in MLIs and Golgi cells. For climbing fiber transduction, AAV1-CaMKIIα-GCaMP6f (University of Pennsylvania) was injected into the inferior olive through a surgical opening in the back of the neck with access to the brainstem through the foramen magnum at the following coordinates from lambda: x = 0.3 mm, y = −4.9 mm, z = −4.6 mm at a depth of 3.6 mm using an approach angle of 62°. Total volumes of injections were ~500 nl. All injection rates were ~25 nl/min.

## Behavior task

Mice were head restrained in a previously described behavior apparatus (*Gaffield et al., 2016*; *Gaffield and Christie, 2017*) by attaching the surgically implanted post to a solid rod that provided stiff resistance to movement while the animal sat in a metal tube (diameter 25.4 mm). Water was delivered through a gavage needle with the end positioned about 2.5 mm from the mouth. All behavior timing was controlled using bControl (Brody Lab; Princeton). Licks were detected using a low-current circuit whereby contact of the tongue with the port produced a small electrical signal that was recorded digitally and could be used to register lick timing against time-series images acquired with 2pLSM. Lick timing was used to calculate lick probabilities within a given time bin. We determined lick rate as the inverse of the average inter-lick interval with the exception of the adjusted lick rate (a measure accounting for the intervals before and after each lick), which was calculated as published previously (*Gaffield and Christie, 2017*). Videography was used to ensure consistent placement of the lick port across all sessions for each mouse.

To provide motivation for participation in the behavior task, mice were maintained on a water-restricted diet (1 ml of water/day) and monitored daily for health. In the first few sessions, mice were familiarized with the apparatus and head restraint, and were trained to lick from the water delivery port by providing free access to water. After mastery, demonstrated by consistent licking, and judged by the uptake of water under high-speed videography, a tone cue (6 kHz) was added signaling water availability. After many trials (100–200 trials/day; 3–5 days), mice learned to lick at the onset of the tone and refrain from licking after fully consuming the dispensed water droplet. Mice typically completed >150 trails until sate. Tone response time was the interval between the end of the tone cue and the first subsequent lick. Licking movement epochs were defined based on whether a lick had been detected within 0.25 s (roughly two lick cycles). Epochs of non-licking were classified based on the absence of licking for >0.75 s.

For experiments in mice expressing the engineered receptor hM4d in MLIs, animals received an intraperitoneal injection of CNO prior to start of the task (45 min; 5 mg/kg; Tocris Bioscience, Bristol, UK; stock solutions were made by dissolving CNO in DMSO to 50 mM). Control measurements in the absence of CNO were made from the same mice on alternate days (>48 hr between sessions).

## In vivo Ca$^{2+}$ imaging

We used 2pLSM to image in vivo Ca$^{2+}$ activity in neurons of the lateral cerebellum. The microscope was purpose built and included resonant scan mirrors (CRS 8K, Cambridge Technologies, Bedford, MA), a low power objective (16X, 0.8 NA water, Olympus, Center Valley, PA), and high-sensitivity PMTs (H10770PA-40, Hamamatsu, Bridgewater, NJ) for both red and green channels producing high-resolution (512 pixels x 512 pixels), high frame rate images (30 frames/s). This rate is faster than the response time of the Ca$^{2+}$ indicators (*Chen et al., 2013*). Imaging was controlled using Scan-Image 2015 software (Vidrio Technologies, Ashburn, VA). In the emission pathway, a 700 nm short-pass filter limited stray excitation light from reaching the PMT detectors. A 570 nm dichroic split the emission light into red and green channels. The green channel also included a 525/50 filter. GCaMP6f was excited at 900 nm (Chameleon Vision S, Coherent, Santa Clara, CA) with <50 mW of power at the objective except for climbing fibers where ~ 100 mW was required. RCaMP2 was excited at 1070 nm (Fidelity 2, Coherent) with <60 mW of power. The same region of cerebellum was imaged across sessions and generally included most of the same PC dendrites although not all dendrites could be automatically re-identified (*Gaffield et al., 2016*).

## Acute brain slice recording

Parasagittal slices of the lateral cerebellum containing lobule Crus II were prepared from mature *Kit:: Cre* mice. Animals were anesthetized with ketamine and xylazine by intraperitoneal injection (100 mg/kg and 10 mg/kg, respectively) and, after opening the chest, were perfused with ice-cold (~4°C) saline through the heart. The cerebellum was then removed by rapid dissection and mounted on an agar block and sectioned in thin slices (200 µm) using a vibraslicer (VT1200S, Leica Biosystems, Buffalo Grove, IL). Sectioning was performed in an ice-cold solution containing (in mM) 87 NaCl, 25 NaHO$_3$, 2.5 KCl, 1.25 NaH$_2$PO$_4$, 7 MgCl$_2$, 0.5 CaCl$_2$, 10 glucose, and 75 sucrose that was continuously bubbled with carbogen gas (95% O$_2$/5% CO$_2$). Once cut, slices were immediately transferred to a holding chamber containing (in mM) 128 NaCl, 26.2 NaHO$_3$, 2.5 KCl, 1 NaH$_2$PO$_4$, 1.5 CaCl$_2$, 1.5 MgCl$_2$ and 11 glucose, maintained at 34°C for 30 min and then at room temperature (~23°C) thereafter. For experiments, acute slices were placed in a recording chamber under a microscope and continuously perfused with an oxygenated saline solution of identical composition to that used for holding after slicing. The concentration of Ca$^{2+}$ was kept low (1.5 mM) to reflect a more physiological-like condition (*Jones and Keep, 1988*) and the solution maintained at a near-body temperature (34–36°C) using an inline heater (TC-344; Warner Instruments). Except where noted, all experiments were performed in the absence of drugs so that both excitatory and inhibitory synaptic transmission were unaffected.

Neurons in Crus II were visually targeted for recording using IR contrast imaging with an upright video microscope (BX51WI; Olympus) and a QIClick CCD Camera (Q-Imaging, Surrey, BC, Canada). PCs and MLIs were easily distinguished based on their location and morphology. Recording pipettes were pulled from thin-walled borosilicate glass (PG 52–165;World Precision Instruments, Sarasota, FL) and filled with a solution containing (in mM) 124 potassium gluconate, 2 KCl, 9 HEPES, 4 MgCl$_2$,

4 NaATP, 3 L-Ascorbic Acid, and 0.5 NaGTP (pH = 7.25). For PC recordings, the $Ca^{2+}$ indicator dye Fluo-5F (200 µM; Life Technologies, Carlsbad, CA) as well as the volume indicator Alexa 594 (60 µM; Life Technologies) were also included in the pipette solution. Cell-attached recordings from MLIs were achieved by forming a loose seal with the patch electrode (~400 MΩ). This prevented dialyzing the cell and changing the intracellular concentration of $Cl^-$ and, hence, the reversal potential for currents generated by *Gt*ACR2.

We used a Multiclamp 700B amplifier (Molecular Devices, Sunnyvale, CA) for electrophysiological recordings. Signals were filtered online at 10 kHz and digitized at 20 kHz with a Digidata 1440 A-D converter (Molecular Devices). For whole-cell recordings, the membrane potential of PCs was maintained −75 mV using constant current injections whereas MLIs were maintained at −70 mV. Current offset was not used during cell-attached recordings of MLIs, allowing these cells to fire spontaneously under their own control, at an unperturbed resting membrane potential. Pipette capacitance was neutralized online and series resistance adjusted using the bridge balance circuitry of the amplifier. Liquid junctional potentials, calculated to be 10 mV, were corrected offline. Climbing fibers were stimulated using bi-polar glass electrodes placed near the axon hillock of the targeted PC. Brief electrical pulses (20 µs; 0.1–1.0 V) were delivered using a stimulation isolation unit (Model DS2A; Digitimer, Ft. Lauderdale, FL). Parallel fibers were also stimulated electrically using an electrode placed in the molecular layer adjacent to the dendritic recording site. The intensity of the stimulus was adjusted to produce PSPs, recorded at the PC soma, of similar amplitude across recordings (0.1–1.0 mV). For conjunctive stimulation experiments, parallel fibers were stimulated in bursts at 100 Hz. Climbing fiber were stimulated 50 ms after the end of parallel fiber tetanus, or at a varying interval where noted. Trials occurred at a relatively low frequency (0.125 Hz). In interleaved trials, either parallel fibers or climbing fibers were stimulated in isolation. This also occurred for trials that included optogenetic actuation of MLIs. Light pulses for optogenetic actuation or inactivation of MLIs activity started 10 ms prior to the beginning of the parallel fiber tetanus and lasted for the duration of the electrical stimulus.

For 2pLSM imaging in slices, we used a commercial scan head (Ultima; Bruker, Billerica, MA) fitted on top of an upright microscope (BX51-WI, Olympus; Tokyo, Japan). The scan head directed laser light (λ = 810 nm) from a mode-locking Ti:sapphire laser (Chameleon Ultra II; Coherent) through a scan lens and pair of galvanometer mirrors (Cambridge Technologies) onto the back aperture of a high-power objective (60X; 1.0 NA). To image $Ca^{2+}$ activity in PCs, indicators dyes were allowed to dialyze for >30 min before starting recordings. Inclusion of the red volume dye allowed for identification of dendrites and spines for subsequent $Ca^{2+}$ activity measurements. However, fluorescence in the red channel was not collected during neural activity measurements because of the interference of light used for the optogenetic stimuli. $Ca^{2+}$ transients were recorded in PC spines using line scans (500 Hz). The PMT used to collect green $Ca^{2+}$-indicator fluorescence was shuttered during blue-light optogenetic stimuli to prevent damage. $Ca^{2+}$ activity was therefore not measured during this period. To facilitate comparison between conditions, a comparable region of the control response was blanked.

For our slice experiments, AAV1-EF1α-Flex(*loxP*)rev-*Gt*ACR2.eYFP (prepared by ViGene) and AAV5-EF1α-Flex(*loxP*)rev-bReaChES-TS-YFP (University of North Carolina) were injected into the lateral cerebellum of *Kit::Cre* mice using a surgical procedure identical to that described above except that the size of the craniotomy was reduced (~0.5 mm diameter). Acute slices were prepared from these mice 7 to 14 days after surgery. *Gt*ACR2 and bReaChES were activated using blue and amber light, respectively, delivered from separate LEDs (M470L3 and M590L3; Thorlabs, Newton, NJ). The emission of the LEDS was combined with a dichroic (T570plxr; Chroma, Bellows Falls, VT) and directed, unfiltered (λ = 461 ± 20 nm and λ = 596 ± 16 nm), into the back epi-port of the microscope. This light was combined into the 2P excitation pathway using a second dichroic (700dcxru; Chroma). LEDs were modulated by separate current controllers (LEDD1B; Thorlabs) using digital commands out of the A-D converter and under computer control from the electrophysiology software (Clampex v10; Molecular Devices).

## Post-hoc histology and confocal imaging

Transgene expression was confirmed by visual inspection of tissue from paraformaldehyde-perfused mice. Following the completion of $Ca^{2+}$ imaging sessions, mice were anesthetized by intraperitoneal injection of ketamine/xylazine (100 mg/kg and 10 mg/kg, respectively) and the chest cavity opened,

exposing the heart. The heart was accessed by a needle and the animal was perfused at 2 ml/min with a 0.1 M phosphate-buffered (PB) solution followed by paraformaldehyde (4% by volume in PB) until the perfusate exiting an opening from the pulmonary artery ran clear. The cerebellum was removed by dissection and sliced into 80 μm sections in cold PB. When necessary, HA immunostaining was used to confirm hM4d expression. Samples were first incubated with anti-HA antibody (#ab9110, Abcam, Cambridge, UK), followed by an Alexa 633 secondary antibody (Thermo Fisher Scientific, Waltham, MA). DAPI (D1306, Thermo Fisher Scientific) counterstaining was used to identify cell locations in some cases. Images were collected on a confocal microscope (LSM 780 Axio Imager 2; Zeiss, Oberkochen, Germany) using 488 nm excitation and 493–598 nm emission for GCaMP6f, 633 nm excitation and 638–747 nm emission for Alexa 633, 405 nm excitation and 410–507 nm emission for DAPI, and 514 nm excitation and 519–620 nm emission for YFP.

## Image analysis

Image analysis of in vivo data was performed blind to the experimental condition. Time-series images of $Ca^{2+}$ activity in neurons expressing genetically encoded $Ca^{2+}$ indicators were aligned using a least-squares algorithm. For dual-color imaging, the translation coordinates from PC images were also used to co-register the corresponding MLI images. Individual PC dendrites or individual climbing fibers were segmented using an independent component analysis algorithm (*Hyvärinen, 1999*; *Gaffield et al., 2016*). Individual MLIs or parallel fibers were identified using hand-drawn ROIs from averaged images. $Ca^{2+}$ events in both PCs and climbing fibers were identified using an inference algorithm (*Vogelstein et al., 2010*); events were identified whether or not they occurred during the decay of other events. However, to avoid potential uncertainties associated with GCaMP6f non-linearity (*Chen et al., 2013*), we performed an initial analysis on well-isolated (non-overlapping) events. For inclusion in this analysis, events must have occurred at least 500 ms from proceeding or following events. This time window allowed for the full decay of climbing fiber-evoked PC $Ca^{2+}$ responses (τ is approximately 150 ms; *Gaffield et al., 2016*). The relative prevalence of isolated events was calculated by counting the number of isolated events, along with the number of times two events occurred within 500 ms, but were isolated by 500 ms from any other events, and the number of times three events occurred within 1000 ms, but were isolated by 500 ms from any other events. A subsequent analysis was performed on overlapping events comprising of two distinct events. In this case, consecutive events were selected that occurred within 150–200 ms of each other; all other events not meeting this criteria were rejected. In a final analysis, we also simply measured the peaks of all algorithmically identified $Ca^{2+}$ responses.

For in vivo measurements of $Ca^{2+}$ events in PCs and climbing fibers, ΔF/F was calculated using a baseline fluorescence period immediately prior to an identified event (~200 ms). For overlapping responses, this baseline period was during the decay of preceding events. For trial-averaged PC $Ca^{2+}$ activity measurements, we calculated ΔF/F for all PC dendrite ROIs using the smallest GCaMP6f-fluorescence values obtained during recordings as the baseline. The average of responses in control was subtracted from that measured in CNO for each mouse before generating an overall average. In bouton measurements from parallel fibers, ΔF/F values were corrected by subtracting the neuropil signal from an area immediately adjacent to each fiber. The MLI-dependent movement (MDM) ratio was defined as the following equation:

$$\left(\frac{\Delta F/F_{CNO}}{\Delta F/F_{Ctrl}}\right)_{movement} \Big/ \left(\frac{\Delta F/F_{CNO}}{\Delta F/F_{Ctrl}}\right)_{no\ movement}$$

ROC analysis involved generating true positive and false positive rates for each threshold in the distribution of fluorescence values from all dendritic pixels reporting $Ca^{2+}$ activity. This was used to generate a ROC curve. The percent of pixels showing a CNO effect was estimated from the area under the ROC curve (*Najafi et al., 2014b*). Analysis of inter-branch climbing fiber $Ca^{2+}$ activity in PCs was performed by selecting two equally sized segments (average area = 190 ± 4 μm²; centers separated by 98 ± 4 μm) from a single dendrite then comparing the amplitudes for each identified event occurring simultaneously at each location. Similar to that of previous work (*Kitamura and Häusser, 2011*), dendritic variability was defined as:

$$\frac{2|A1 - A2|}{A1 + A2}$$

Where A1 and A2 are the peak fluorescence amplitudes for branch 1 and branch 2, respectively.

For ex vivo $Ca^{2+}$ imaging experiments, fluorescence changes in PC dendrites were quantified as $\Delta F/F$ (average ~10 trials per condition). The peak climbing fiber-evoked $Ca^{2+}$ transient was determined from an exponential fit of the fluorescence decay immediately following the electrical stimulus.

All image analysis was performed with Matlab (Mathworks, Natick, MA) or ImageJ (NIH). AxoGraph (Axograph) was used to analyze electrophysiological data. Additional calculations and plotting was performed with Excel (Microsoft, Redmond, WA) and Prism (GraphPad, La Jolla, CA). In figures, error bars indicate SEM. For data contained within the figure set where individual measurements are not already shown please see *Figure 1—source data 1*, *Figure 1—figure supplement 1—source data 1*, *Figure 1—figure supplement 2—source data 1*, *Figure 2—source data 1*, *Figure 2—figure supplement 1—source data 1*, *Figure 2—figure supplement 2—source data 1*, *Figure 2—figure supplement 3—source data 1*, *Figure 4—source data 1*, and *Figure 5—figure supplement 2—source data 1*.

## Acknowledgements

We thank the members of the Christie Lab for their helpful discussion during preparation of this manuscript. We are especially grateful to the GENIE program and the Janelia Research Campus (Drs. Jayaraman, Kerr, Kim, Looger, and Svoboda) for generously making GCaMP6f widely available to researchers and Dr. Bito (University of Tokyo) for use of RCaMP2. This work was supported by the Max Planck Society, the Max Planck Florida Institute for Neuroscience, and National Institutes of Health Grant NS083894 (JMC).

## Additional information

### Funding

| Funder | Grant reference number | Author |
| --- | --- | --- |
| National Institutes of Health | NS083894 | Jason Christie |
| Max-Planck-Gesellschaft | | Jason Christie |
| Max Planck Florida Institute for Neuroscience | | Jason Christie |

The funders had no role in study design, data collection and interpretation, or the decision to submit the work for publication.

### Author contributions

Michael A Gaffield, Conceptualization, Formal analysis, Investigation, Methodology, Writing—original draft; Matthew J M Rowan, Formal analysis, Investigation, Methodology, Writing—review and editing; Samantha B Amat, Methodology, Writing—review and editing; Hirokazu Hirai, Resources, Writing—review and editing; Jason M Christie, Conceptualization, Funding acquisition, Methodology, Writing—original draft, Project administration

### Author ORCIDs

Jason M Christie (ID) https://orcid.org/0000-0003-0276-2554

### Ethics

Animal experimentation: Animal procedures were conducted using protocol 15-205 approved by the Institutional Animal Care and Use Committee (IACUC) at Max Planck Florida Institute for Neuroscience.

#### Decision letter and Author response
Decision letter https://doi.org/10.7554/eLife.36246.029
Author response https://doi.org/10.7554/eLife.36246.030

## Additional files

### Supplementary files
• Transparent reporting form
DOI: https://doi.org/10.7554/eLife.36246.027

### Data availability
All data are included in the manuscript or the source data files.

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
