## [Decision Letter]

Thank you for submitting your article "Inhibition gates instructional Ca^2+^ signaling in Purkinje cell dendrites during movement" for consideration by *eLife*. Your article has been reviewed by three peer reviewers, including Jennifer L Raymond as the Reviewing Editor and Reviewer #3, and the evaluation has been overseen by Eve Marder as the Senior Editor. The following individual involved in review of your submission has agreed to reveal their identity: Martijn Schonewille (Reviewer #1).

The reviewers have discussed the reviews with one another and the Reviewing Editor has drafted this decision to help you prepare a revised submission.

Summary:

This manuscript by Gaffield and colleagues described an elegant series of experiments investigating the effects of excitatory and inhibitory synaptic inputs to the Purkinje cells climbing fiber-evoked calcium signals. Historically, the climbing fibers were thought to elicit a reliable, all-or-none (binary) calcium response in Purkinje cells, which triggered associative LTD in coactive parallel fibers. However, a number of recent in vitro and in vivo studies have demonstrated that the climbing fiber-elicited calcium response is graded, and that variations in the amplitude of the calcium response can modify the probability that associative plasticity is induced. The current manuscript extends this work by identifying inhibition from the molecular layer interneurons (MLIs) as a signal that can regulate the calcium response elicited by the climbing fibers.

The authors employ a powerful combination of complementary in vivo and in vitro calcium imaging, optogenetic and pharmacogenetic manipulation, and slice physiology. They show that the amplitude of calcium transients during a well-learned behavior (licking), which increases both MLI activity and the rate of these transients (Figure 1), is similar to calcium transients in the absence of movement. Chemogenetic inactivation of MLIs, however, unmasks a widespread, movement-specific increase in transient amplitude (Figure 2, Figure 3). This increase appears to be postsynaptic, as direct imaging of climbing fibers fails to reveal an influence of the ligand on presynaptic calcium preceding or during movement (Figure 4). Optogenetic suppression of MLIs in acute slices enhances parallel fiber (PF) postsynaptic potentials at the Purkinje cell soma and induces a supralinear dendritic calcium response to paired parallel and climbing fiber stimulation. Finally, the authors demonstrate that long trains of PF stimulation can induce a similar supralinear calcium response, but that this interaction is erased by optogenetic activation of MLIs.

The work is of high potential impact. The results extend our understanding of events that can gate or modulate the climbing fiber's impact on their Purkinje cell targets, which is thought to provide instructive signals that guide the induction of cerebellum-dependent learning. The experiments are well-designed, the claims are clearly explained, and the data support the claims. However, the reviewers agreed that some issues should be addressed by the authors.

Essential revisions:

1) Because long trains of PF stimulation can induce a supralinear PF+CF effect on calcium, the authors should rule out the possibility that pharmo- and opto-genetic perturbations drives an increase in PF activity (e.g. by increasing the frequency of some possibly unobserved movement, or via inhibition of Golgi cells, which also have some expression in the c-kit line). The most direct way to achieve this would be to perform calcium imaging in granule cells and check that activity is similar between treatment and sham conditions. It may also be helpful to provide a more thorough analysis of the effects of the MLI manipulations on behavior. Is enhanced calcium still observed with CNO if the analysis is limited to CNO and control trials with the behavior as closely matched as possible (subsample Figure 2C data to control for licking rate)?

2) More thorough analysis of the data, and more complete presentation of the results are needed throughout the manuscript.

a) In the analysis of the in vivo calcium events, the authors focus on the unitary calcium responses, and exclude the multipeaked responses from most pf the analyses. The description of how the unitary events were identified is scanty (subsection “Image Analysis”). The authors should provide more information about how they did the analysis and provide an analysis of the robustness of the results to their specific choices for the analysis. What does the distribution of all (single and multipeaked) calcium event amplitudes look like? What is the distribution of single, double, triple peaked events? Does the latter differ during cued licking behavior vs. spontaneous? The example trace in Figure 1C suggests there is surprisingly large variation in the size of unitary events (a ~3-fold difference in the peak with the green check mark vs. the first peak in the multiple event)-this makes me wonder how many of the unitary events selected for analysis are actually composed of two smaller calcium events occurring close together in time. What is the temporal resolution for discriminating one vs. two closely spaced events? A more careful consideration of how the choice of analysis could affect the results and the interpretation of the results is needed.

b) The authors show in Figure 3C that ~63% of the dendritic area shows enhanced Ca^2+^ signaling after MLI disinhibition. If I understand correctly the figure also indicates that the other ~37% has a substantial decrease of the Ca^2+^. This should be addressed, also in relation to the chance of random changes in Ca^2+^ signaling (is the 67% statistically significant from random 50%). Similarly, the remark about the 'arbor-wide' influence should be re-worded to more careful terms.

c) For Figure 5D and Figure 6, it would be very helpful if the evoked PF and CF transients were also shown individually, before the summed transient is shown. Also, please state clearly if the summed response for the suppressed inhibition is based on the PF with or without inhibition.

d) Figure 6G and associated text suggest that the MLIs exert a graded effect on the climbing fiber-elicited calcium response. Is there truly a graded effect on amplitude, or might it be an effect on the probability of an additive vs. supralinear event of a more uniform amplitude? Additional analysis could distinguish these two equally interesting possibilities.

e) The point of Figure 1 is that the amplitude of the calcium events is the same during movement or no movement. However, in Figure 2F, this does not seem to be the case for the hM4d animals, even in the absence of CNO. If the calcium sensing is nonlinear, this could potentially influence what is measured with CNO. This possibility should be considered.

f) There are some places where individual examples are shown, but no group data, for example, Figure 4E.

g) There are some places where results are presented without statistics. For example, is there is a significant difference between movement vs. no movement and between CNO vs. control in Figure 3FG? Also, the error bars on the CNO-control PC data for simultaneous PC/MLI imaging (Figure 2G, red trace) are very large. Please provide the same statistical analyses for these data as were used for the data in Figure 2F to justify the claim that the difference is positive. Also, please provide the same analysis for the climbing fiber imaging data in Figure 4G, where the difference appears to be negative, but is asserted to not differ from zero.

h) To support the claim that peak PC ΔF/F does not co-vary with MLI activity (Figure 1G), please plot compute the correlation between the peak amplitude and MLI activity for all dendrites and complex spike event, and display a scatterplot of these values.

i) A video of mouse behavior and two-color imaging (Figure 1) would be useful.

j) Figure 5D shows that MLI inactivation induces a supralinear interaction between PF and CF stimulation on dendritic calcium. Is the same true of the somatic membrane potential? Please overlay plots of the PF+CF and PF+CF (sum) for control and light conditions (similarly to Figure 5D) in Figure 5F, and make a similar plot for the ephys data in the long pulse train experiments (Figure 6E).

k) Was there any correlation between calcium transient amplitude and behavior? For instance, were transients larger on trials with longer or mistimed licking bouts? Similarly, did the correlation between PC calcium and MLI calcium change with any aspect of behavior? Although the authors have addressed some of these questions in previous work, it would be helpful to have this information for the current manipulations and data.

l) In general, the figure legends and text describing the figures should be edited to be more precise and make it easier for the reader to understand what is in the figure. For example, in Figure 2G and associated text, I am guessing that the "totality of all calcium responses" includes the multipeaked as well as the unitary responses, but after considerable effort, I am still not sure. Another example, it is not clear whether the averaged calcium activity in Figure 1F is from one mouse or all mice. Please carefully review all figure legends to make sure that the reader is provided with sufficient information about what is in each figure panel.

m) Please provide time units for the event probability plots to allow comparison across figures, and with previously published climbing fiber firing rates. I would expect the probabilities in Figure 1E and Figure 4E to be similar, but they differ by twofold. How does this compare with the typical climbing fiber rate of 1 Hz?

3) The current results would have more impact if more effectively framed in the context of what is already known about variations in climbing fiber-triggered responses in the Purkinje cells, how the current work extends what is known, the conclusions that can be drawn, and any caveats.

a) Previous work has associated the calcium transients with plasticity, however, the reviewers thought there was too much emphasis on plasticity (using "instructional Ca^2+^ signaling" in the Title and last sentence of the Abstract and "circuit modifications" in the first sentence of the Abstract), given that plasticity is not tested in the in vitro or in vivo experiments.

b) Najafi et al., have reported that climbing fiber-associated calcium responses are enhanced when they occur during a learning task as compared with the spontaneous calcium events, which are presumably due to spontaneous climbing fiber spiking. In contrast, the current results indicate that the calcium responses during the well-learned lick task are the same as the spontaneous calcium events. The authors cite Najafi, but don't directly compare the two results so that the reader can effectively appreciate the new finding that the calcium responses are not always enhanced in the behavioral context. This would better frame the current Results section than the current statement in the Introduction "whether CF Ca^2+^ signals in PC dendrites are augmented by preceding PF activity in vivo is unclear" and would explain why the results in Figure 1D are described as "Surprising".

c) More discussion of the cued licking task, and the role of the Ca^2+^ signal in Purkinje cells during the movement in the lick task is needed. From the example mouse data in Figure 1B, it is difficult to tell how much of the behavioral response and neural activity is learned (a conditioned response) versus an unconditioned response to the water — it looks like the majority of the neural and behavioral response could be unconditioned. Which components of the lick task are cerebellum dependent-acquisition of learning? expression of the condition response? performance of unconditioned licks? Discussion of these issues would help to clarify the potential role of the calicum transients in the behavior.

d) In Figure 5 and Figure 6, might the lack of supralinear calcium responses result from the use of a suboptimal parallel fiber-climbing fiber pairing interval?

[Editors' note: further revisions were requested prior to acceptance, as described below.]

Thank you for submitting your article "Inhibition gates supralinear Ca^2+^ signaling in Purkinje cell dendrites during practiced movements" for consideration by *eLife*. Your article has been reviewed by three peer reviewers, including Jennifer L Raymond as the Reviewing Editor and Reviewer #3, and the evaluation has been overseen by Eve Marder as the Senior Editor. The following individual involved in review of your submission has agreed to reveal their identity: Martijn Schonewille (Reviewer #1).

The reviewers have discussed the reviews with one another and the Reviewing Editor has drafted this decision to help you prepare a revised submission.

Summary:

This manuscript provides evidence that the molecular layer interneurons in the cerebellar cortex modulate the climbing fiber-triggered calcium responses in Purkinje cells. Historically, the climbing fibers were thought to elicit a reliable, all-or-none (binary) calcium response in Purkinje cells, which triggered associative LTD in coactive parallel fibers. However, a number of recent in vitro and in vivo studies have demonstrated that the climbing fiber-elicited calcium response is graded, and that variations in the amplitude of the calcium response can modify the probability that associative plasticity is induced. The current manuscript extends this work by identifying inhibition from the molecular layer interneurons (MLIs) as a signal that can regulate the calcium response elicited by the climbing fibers. The authors employ a powerful combination of complementary in vivo and in vitro calcium imaging, optogenetic and pharmacogenetic manipulation, and slice physiology. In response to the previous review, the authors have conducted a number of additional analyses, which increase the rigor and strengthen the conclusions. In addition, they included new data from experiments imaging the parallel fiber axons during their cued licking task. Overall, this study represents a significant advance in understanding how the cerebellum implements learning, in particular, the events governing the induction of associative LTD at the parallel fiber-Purkinje cell synapse. The reviewers did not think additional experiments were necessary for this to make a valuable contribution, however, they felt that some additional revisions of the text were needed to better acknowledge and discuss some of the limitations of the current study that were raised in the previous review, and which were not fully addressed by the additional analyses and experiments provided in the revised manuscript.

Essential revisions:

1) The manuscript provides convergent evidence that inhibition from the molecular layer interneurons (MLIs) can gate supralinear calcium responses to combined parallel fiber and climbing fiber activity. This is likely correct. However, it is also difficult to rule out the possibility that in vivo, the chemogenetic suppression of MLIs could have resulted in changes (increases) in granule cell/parallel activity, which could contribute to the enhanced calcium responses. The authors need to be more forthcoming and explicit in the manuscript about acknowledging this possibility and the limited extent to which their evidence addresses it.

Their argument that argument that parallel fiber activity is not altered by suppression manipulation of MLI activity hinges on:

a) new data showing that in the absence of MLI manipulation, calcium imaging-based measurement of parallel fiber activity closely tracks MLI activity (Figure Figure 1—figure supplement 1) and an assertion in their point-by-point response that "unilateral chemogenetic disinhibition of MLIs in Crus II did not affect the population response of MLIs (data not shown). There are some concerns with this argument. First, just because the activity of two populations of neurons is similar under one measurement condition, it does not mean that it will be similar under all conditions, such as experimental manipulation of inhibition. Second, the whole point of the chemogenetic manipulation of MLIs is to suppress their activity, so why does it not affect the population response? Are we missing something? Third, the Kit promoter has some expression in Golgi cells, which directly inhibit the granule cells-this needs to be acknowledged.

b) Lack of change in the behavior. The additional data provided in Figure 2—figure supplement 3 is helpful, but does not fully address the possibility of other, unmeasured behavior differences (lateral deviation of the tongue, other orofacial movements), which is why the reviewers had asked for sample videos with and without chemogenetic suppression. Also, the lack of behavioral difference does not rule out a difference in the granule cells that does not affect the behavior, but could contribute to calcium transients in the Purkinje cells.

In the absence of more direct recordings from the granule cells or parallel fibers comparing responses in the presence or absence of the disinhibition manipulation, we would be satisfied with a more thorough acknowledgement and discussion of the above caveats in the manuscript.

2) Figure 1—figure supplement 1B and 1E show that PF activity is well correlated with licking behavior after the cue. However, it is not very clear in Figure 1—figure supplement 1D. In particular, fluorescence signals begin to increase mostly before licking starts in the B3 bouton. It is therefore important to show PF and MLI activity not only when the cue induces licking but also fails to induce licking as described above. In addition, please indicate in Figure 1—figure supplement 1D when the cue was presented. Otherwise, readers cannot tell which licking bouts are learned behavior.

3) In the first round of review, one of the reviewers asked more discussion of the cued licking task, for example, which components of the lick task are cerebellum dependent-acquisition of learning. The authors address the comment, but their rebuttal letter has addressed the comment better than the manuscript itself. It is important to let readers know that it is currently unclear (1) the role of climbing fibers in the cued licking task, and (2) which aspects of this learning task are regulated by the cerebellum. These limitations do not diminish the value of this study.

---

## [Author Response]

Essential revisions:1) Because long trains of PF stimulation can induce a supralinear PF+CF effect on calcium, the authors should rule out the possibility that pharmo- and opto-genetic perturbations drives an increase in PF activity (e.g. by increasing the frequency of some possibly unobserved movement, or via inhibition of Golgi cells, which also have some expression in the c-kit line). The most direct way to achieve this would be to perform calcium imaging in granule cells and check that activity is similar between treatment and sham conditions. It may also be helpful to provide a more thorough analysis of the effects of the MLI manipulations on behavior. Is enhanced calcium still observed with CNO if the analysis is limited to CNO and control trials with the behavior as closely matched as possible (subsample Figure 2C data to control for licking rate)?

We have collected a new set of data using GCaMP6f to measure the activity of parallel fiber axons during our cued-licking task. Parallel fibers were robustly activated during the consumption of water and were relatively inactive in the absence of licking (Figure 1—figure supplement 1). Parallel fiber activity also closely tracked licking kinematics. Interestingly, the movement-induced activation of MLIs was essentially an exact match of that of the average response of the parallel fibers in the same region. This indicates that MLI activity measurements can be used to infer the overall activity of the granule cell population. Unilateral chemogenetic disinhibition of MLIs in Crus II did not affect the population response of MLIs, measured using our Ca^2+^ imaging approach. Thus, by inference, disinhibition did not affect the overall level of granule cell activity. This rules out the possibility that enhanced Ca^2+^ signaling in PCs was the result of a CNO-induced change in the presynaptic activity level of granule cells.

Addressing the specific question of parallel fiber signaling onto Purkinje cell dendrites with and without MLI block is not technically feasible in awake behaving animals at this time. We are therefore left with relying on the reduced preparation experiments where we show that shifting the balance between parallel fiber excitation and MLI inhibition of PCs can likewise affect PC Ca^2+^ activity. These points are now emphasized in the revision.

We apologize for not being forward in our reporting of potential motor changes with chemogenetic suppression of MLI activity. We actually went out of our way to identify conditions that were not conducive to affect behavior as this could complicate the interpretation of our results. We now present our analysis showing the absence of changes in average licking performance following unilateral suppression of MLI activity in the small area of left Crus II under our imaging window (Figure 2—figure supplement 3). We do not think this result contradicts our previous work where we reported a small, but significant reduction in lick rate with MLI activity suppression by hM4d activation (Gaffield et al., 2017). This is because in this prior work, we inhibited MLI activity bilaterally in large swaths of both left and right Crus II, a much more robust perturbation of the MLI population controlling orofacial function. At the reviewers’ request, we examined PC Ca^2+^ activity in a subset of mice showing the smallest deviations in lick rate with chemogenetic disinhibition, compared to control. MLI activity suppression continued to enhance climbing fiber-evoked Ca^2+^ events in PCs from these mice (Figure 2—figure supplement 3). Moreover, limited behavioral differences indicate that granule cell activity (which tracks behavior) is also unlikely to change with chemogenetic inhibition of MLIs, which further supports our conclusion that the observed alteration of PC Ca^2+^ signaling by chemogenetics is not due to activity changes in granule cells.

2) More thorough analysis of the data, and more complete presentation of the results are needed throughout the manuscript.

We have made extensive new additions to the manuscript including many new supplemental figures to address each of the reviewers’ concerns.

*a) In the analysis of the* in vivo *calcium events, the authors focus on the unitary calcium responses, and exclude the multipeaked responses from most pf the analyses. The description of how the unitary events were identified is scanty (subsection “Image Analysis”). The authors should provide more information about how they did the analysis and provide an analysis of the robustness of the results to their specific choices for the analysis. What does the distribution of all (single and multipeaked) calcium event amplitudes look like? What is the distribution of single, double, triple peaked events? Does the latter differ during cued licking behavior vs. spontaneous? The example trace in Figure 1C suggests there is surprisingly large variation in the size of unitary events (a ~3-fold difference in the peak with the green check mark vs. the first peak in the multiple event)-this makes me wonder how many of the unitary events selected for analysis are actually composed of two smaller calcium events occurring close together in time. What is the temporal resolution for discriminating one vs. two closely spaced events? A more careful consideration of how the choice of analysis could affect the results and the interpretation of the results is needed.*

We apologize for the inadequacy of our description of unitary event selection for analysis of PC dendritic Ca^2+^ signals. Specifically, we used a previously published inference algorithm (Vogelstein et al., 2010) to identify all climbing fiber-evoked Ca^2+^ events in PCs, whether or not they occurred during the decay of preceding events. The reviewers refer to these overlapping events as “multi-peaked”. We are hesitant to use this terminology as it seems to imply some sort of physiological relevance rather than simply reflecting that climbing fiber-evoked events can occur in succession and the decay time of GCaMP6f is relatively slow (τ is approx. 150 ms for PCs [Gaffield et al., 2016]). We rejected overlapping dendritic Ca^2+^ events from our initial analysis because of possible uncertainties of GCaMP6f nonlinearity. Therefore, our criteria for inclusion was that an event occurred at least 500 ms from proceeding or following events, a window that allows for GCaMP6f fluorescence to decay back to baseline levels (as shown in Figure 1D). We have now included a better description of our selection criteria and rationale in the Materials and methods section.

In response to the reviewers urging, we now report the distribution of single vs. overlapping events. Isolated, single events comprise the majority of the response population; this makes sense because the average Ca^2+^ event rate in PCs is low, about 1.5 Hz. We found that these distributions don’t change during motor behavior (Figure 1—figure supplement 2).

A major finding of our report is that with chemogenetic suppression of MLI activity, Ca^2+^ event amplitudes increased during movement. This is a robust result. New analysis shows that this holds true not only for discrete single events, but also for all other events including overlapping responses (Figure 2—figure supplement 1). Regarding the discrimination of overlapping Ca^2+^ events, our imaging rate is 30 Hz (~33 ms); this defines a theoretical temporal resolution by which we could distinguish two closely spaced dendritic events. That said, we don’t believe that unresolved, closely-spaced responses appearing as single events accounted for the difference in response amplitude with chemogenetic disinhibition. If so, this would have been reflected in the presynaptic activity of climbing fibers. Because Ca^2+^ event amplitudes (and frequency) did not change in climbing fibers with molecular layer disinhibition, this argues strongly against this possibility. We now state this in the text.

Like the reviewers, we are fascinated by the variability in the amplitudes of PC dendritic Ca^2+^ events. Identifying the mechanistic underpinnings of this variability is a major goal of ours. That similar variability is apparent in the amplitudes of Ca^2+^ events in climbing fibers indicates the PC dendrites are simply integrating the level of presynaptic activity from these inputs. We hypothesize that MLI-mediated gating of dendritic nonlinearities allow for the faithful conversion of presynaptic climbing fiber activity into postsynaptic Ca^2+^ signals in PCs. However, this investigation is beyond the scope of the current report.

b) The authors show in Figure 3C that ~63% of the dendritic area shows enhanced Ca^2+^ signaling after MLI disinhibition. If I understand correctly the figure also indicates that the other ~37% has a substantial decrease of the Ca^2+^. This should be addressed, also in relation to the chance of random changes in Ca^2+^ signaling (is the 67% statistically significant from random 50%). Similarly, the remark about the 'arbor-wide' influence should be re-worded to more careful terms.

To address this concern, we have changed our analysis approach and now include use of receiver operating characteristic (ROC) curves to diagnose the probability of disinhibition-induced shifts in Ca^2+^ activity across all pixels of identified PC dendrites (i.e., area under the curve). This analysis indicates a widespread effect (Figure 3C). In addition, we have re-worded the Results section, taking caution about the use of the phrase ‘arbor-wide influence’.

c) For Figure 5D and Figure 6, it would be very helpful if the evoked PF and CF transients were also shown individually, before the summed transient is shown. Also, please state clearly if the summed response for the suppressed inhibition is based on the PF with or without inhibition.

We now show both the individual, parallel fiber- and climbing fiber-evoked Ca^2+^ transients in Figure 5D (the parallel fiber response is always quite small or negligible for all recordings). This should help readers understand our approach for quantifying supralinearity in dendritic PC Ca^2+^ signaling more easily. The same method applied for dendritic Ca^2+^ activity measurements in Figure 6. We now state, in the accompanying legend of Figure 5, that the summed response for the suppressed condition is based on a PF stimulus without feed-forward inhibition (MLI activity during the parallel fiber tetanus was eliminated by *Gt*ACR2).

d) Figure 6G and associated text suggest that the MLIs exert a graded effect on the climbing fiber-elicited calcium response. Is there truly a graded effect on amplitude, or might it be an effect on the probability of an additive vs. supralinear event of a more uniform amplitude? Additional analysis could distinguish these two equally interesting possibilities.

Single-trial analysis of climbing fiber-evoked Ca^2+^ transients could distinguish these possibilities. However, the feasibility of such analysis has not been established for PCs, at least to our knowledge. This is because single trial analysis is made difficult due of the low signal-to-noise level of climbing fiber-evoked responses. We could change our recording parameters in an attempt to help improve the quality of evoked Ca^2+^ signals (e.g., increasing the concentration of external Ca^2+^ or the number of conjunctive climbing fiber stimuli, etc.). This would be very laborious and time consuming. In the end, parsing which mechanism is responsible for the observed effect would not change the overall interpretation of our results as both possibilities are interesting and relevant. Therefore, we have removed the term “graded” from the manuscript in favor of simply stating that the effect of parallel fiber excitation on climbing fiber-evoked Ca^2+^ signaling in PCs was dependent on the level of MLI activity.

e) The point of Figure 1 is that the amplitude of the calcium events is the same during movement or no movement. However, in Figure 2F, this does not seem to be the case for the hM4d animals, even in the absence of CNO. If the calcium sensing is nonlinear, this could potentially influence what is measured with CNO. This possibility should be considered.

The reviewers point to a trend towards a larger Ca^2+^ response with movement, compared to events in the absence of movement, in hM4D-expressing mice. However this difference is not significant. We now state this in legend of Figure 2. The major concern with uncertainties regarding GCaMP6f nonlinearity is whether there is a faithful reporting of the relative change in amplitudes (including the absence of change due to saturation). This is compounded for responses that occur in the decay of others because Ca^2+^ unbinding isn’t yet complete. This is a major reason why we avoided including such events in our initial analysis (we now include such events in a second analysis at the reviewers’ request). Heeding this concern, we were also hesitant about making significant conclusions based on how much change we observed in the disinhibited condition. Thus, we don’t believe that enhancement of Ca^2+^ signaling by GCaMP6f can be explained by a nonlinearity, although the absence of effect may warrant caution. In response to the reviewers, we now state in the text that GCaMP6f is known to be nonlinear but, based on previous reports, is sensitive enough to report small changes in climbing fiber-evoked PC Ca^2+^ signaling to different behavioral stimuli. This point is quite moot later in the manuscript when we show that enhanced Ca^2+^ signaling can be uncovered with chemogenetic-induced, molecular layer disinhibition.

f) There are some places where individual examples are shown, but no group data, for example, Figure 4E.

We have changed Figure 4E to include group data; we can find no other instances where group data are not reported either in the text or in a separate Figure panel.

g) There are some places where results are presented without statistics. For example, is there is a significant difference between movement vs. no movement and between CNO vs. control in Figure 3FG? Also, the error bars on the CNO-control PC data for simultaneous PC/MLI imaging (Figure 2G, red trace) are very large. Please provide the same statistical analyses for these data as were used for the data in Figure 2F to justify the claim that the difference is positive. Also, please provide the same analysis for the climbing fiber imaging data in Figure 4G, where the difference appears to be negative, but is asserted to not differ from zero.

The statistical comparisons for the data shown in Figure 3F and 3G are presented in Figure 3H (in this comparison, mean variability of PC Ca^2+^ activity was significantly different following chemogenetic suppression of MLIs). We apologize for not pointing out this figure panel in our original submission. This has now been corrected.

For activity measurements in climbing fibers, we compared the peak amplitudes of averaged Ca^2+^ events collected in control and during disinhibition of the molecular layer (Figure 4G). The difference trace, like those shown in Figure 2C-2E, are for the reader’s convenience. The statistical test for this comparison is reported in the text; it was non-significant (P = 0.66; Student’s t-test).

Regarding our results examining trial-averaged PC Ca^2+^ activity during cued licking, we show the subtracted difference between responses measured in control and with molecular layer disinhibition (Figure 2G, top panel, red trace). We chose to present the subtracted difference because it facilitated comparison to the activity plot of MLIs (i.e., the change in PC activity with disinhibition closely corresponded to the time period when MLIs were normally activated by the behavior). This was also for the reader’s convenience but, as requested, we performed a statistical analysis of this dataset (inset of Figure 2G). Based on the reviewers’ comments, we believe presenting the data as we did may have caused confusion. Therefore, we now show responses of trial-average PC activity in the two conditions used to generate the difference plot (control and with chemogenetic MLI activity suppression) as well as an accompanying statistical analysis of these data (Figure 2—figure supplement 2).

h) To support the claim that peak PC ΔF/F does not co-vary with MLI activity (Figure 1G), please plot compute the correlation between the peak amplitude and MLI activity for all dendrites and complex spike event, and display a scatterplot of these values.

As shown in our analysis of mean activity (Figure 1G), there is no relationship between the amplitude of climbing fiber-evoked dendritic Ca^2+^ events in PCs and the corresponding level of MLI activation. However, in response to the reviewers, we have provided scatterplots from three representative mice showing all PC dendritic events and the corresponding level of activity in surrounding MLIs (Figure 1—figure supplement 3).No relationship was found.

i) A video of mouse behavior and two-color imaging (Figure 1) would be useful.

We show a plot of Ca^2+^ activity measured simultaneously in PCs and MLIs using a dual-color, Ca^2+^ indicator imaging approach. The average responses are demarcated based on their correspondence to cell type and aligned to the onset of behavior (licking). We are not sure how the addition of a video would help clarify our approach or provide any additional insight into this result to readers.

j) Figure 5D shows that MLI inactivation induces a supralinear interaction between PF and CF stimulation on dendritic calcium. Is the same true of the somatic membrane potential? Please overlay plots of the PF+CF and PF+CF (sum) for control and light conditions (similarly to Figure 5D) in Figure 5F, and make a similar plot for the ephys data in the long pulse train experiments (Figure 6E).

Hyperpolarization of the membrane potential by somatic current injection diminishes climbing fiber-evoked responses in PC dendrites including supralinear Ca^2+^ signaling (Wang et al., 2000, Kitamura et al., 2011, and Otsu et al., 2014). This is likely mediated through the passive spread of voltage into the dendritic compartment, preventing parallel fiber-mediated inactivation of Kv4 currents (Otsu et al., 2014). It follows then that GABA_A_R-mediated hyperpolarization, elicited by MLI activity, is sufficient to prevent the supralinear interaction of parallel fibers and climbing fibers in PC dendrites, as observed in our experiments. At the reviewers’ request, traces of complex spikes with and without preceding parallel fiber stimulation are shown superimposed in Figure 5F for both control and the responses with optogenetic suppression of MLIs. For the data presented in Figure 6, these are included as a supplement additional figure (Figure 6—figure supplement 1).

k) Was there any correlation between calcium transient amplitude and behavior? For instance, were transients larger on trials with longer or mistimed licking bouts? Similarly, did the correlation between PC calcium and MLI calcium change with any aspect of behavior? Although the authors have addressed some of these questions in previous work, it would be helpful to have this information for the current manipulations and data.

We did not observe significant changes to average licking behavior with unilateral chemogenetic suppression of MLI activity in left Crus II. This is now reported in the manuscript (Figure 2—figure supplement 3). A more in depth examination of deviant licking, perhaps reflecting motor errors that lead to learning, was not pursued. Regarding motor errors and climbing fiber-evoked signals, it is certainly a future interest of ours to understand the behavioral conditions conducive to producing PC supralinear Ca^2+^ responses. To address this point carefully, will require more exacting control of the movement task as well as additional tools to measure and manipulate neural activity in the granule cell and molecular layers. Thus, this is beyond the scope of the current study. As the reviewers point out, we have previously reported that the rate of climbing fiber-evoked Ca^2+^ events in PC dendrites increases at the onset of licking while MLI activity increases and decreases with changes in licking rate during water consumption. This is now discussed in greater detail.

l) In general, the figure legends and text describing the figures should be edited to be more precise and make it easier for the reader to understand what is in the figure. For example, in Figure 2G and associated text, I am guessing that the "totality of all calcium responses" includes the multipeaked as well as the unitary responses, but after considerable effort, I am still not sure. Another example, it is not clear whether the averaged calcium activity in Figure 1F is from one mouse or all mice. Please carefully review all figure legends to make sure that the reader is provided with sufficient information about what is in each figure panel.

The “totality of all calcium responses” was meant to indicate that we simply averaged all Ca^2+^ activity (including overlapping events spaced closely in time) in all identified PCs across all trials. We have attempted to improve the clarity of our language regarding this dataset and have added a new figure to illustrate this result more directly (Figure 2— figure supplement 2). We now report numbers for Figure 1F in the legend.

m) Please provide time units for the event probability plots to allow comparison across figures, and with previously published climbing fiber firing rates. I would expect the probabilities in Figure 1E and Figure 4E to be similar, but they differ by twofold. How does this compare with the typical climbing fiber rate of 1 Hz?

We changed the plots to include time units (rate/frequency). We show that the average rate of Ca^2+^ events is the same for both climbing fibers and PCs. The increase in peak Ca^2+^ event rates at the initiation of licking is also not significant between climbing fibers and PCs (there is large amount of variability in the climbing fiber response); we don’t report this because our preference is to more carefully compare climbing fibers and PCs using simultaneous dual-color imaging of pre- and post-synaptic activity in the same mouse which we hope to publish in the near future. The rates of Ca^2+^ events detected in PCs in our experiments closely matches that of previously published values. This is now stated in the text with accompanying references. To our knowledge, we are unaware of anyone else who has measured activity rates directly from climbing fibers, so we are proud to say that this is a first!

3) The current results would have more impact if more effectively framed in the context of what is already known about variations in climbing fiber-triggered responses in the Purkinje cells, how the current work extends what is known, the conclusions that can be drawn, and any caveats.

We thank the reviewers for highlighting areas of our manuscript where we could include clarification to increase the impact of our work.

*a) Previous work has associated the calcium transients with plasticity, however, the reviewers thought there was too much emphasis on plasticity (using "instructional Ca^2+^ signaling" in the Title and last sentence of the Abstract and "circuit modifications" in the first sentence of the Abstract), given that plasticity is not tested in the* in vitro *or* in vivo *experiments.*

As requested, we have removed most of these phrases from the Title and Abstract. However, the first sentence of the Abstract simply describes what the cerebellum does and emphasizes the motivation for understanding dendritic Ca^2+^ signaling in PCs. Similarly in the Introduction, our use of terms like “circuit alterations” and “plasticity” is necessary to put our experiments in context. Therefore we kept these words in the first two paragraphs while removing them from the third as this was merely speculative.

*b) Najafi et al. have reported that climbing fiber-associated calcium responses are enhanced when they occur during a learning task as compared with the spontaneous calcium events, which are presumably due to spontaneous climbing fiber spiking. In contrast, the current results indicate that the calcium responses during the well-learned lick task are the same as the spontaneous calcium events. The authors cite Najafi, but don't directly compare the two results so that the reader can effectively appreciate the new finding that the calcium responses are not always enhanced in the behavioral context. This would better frame the current Results section than the current statement in the Introduction "whether CF Ca^2+^ signals in PC dendrites are augmented by preceding PF activity* in vivo *is unclear" and would explain why the results in Figure 1D are described as "Surprising".*

We thank the reviewers for pointing this out. We now emphasize the novelty of our findings regarding that lack of behavior-related augmentation of Ca^2+^ signaling in PCs. A detailed comparison of the work by Najafi et al., contrasting their findings to ours, is also provided.

c) More discussion of the cued licking task, and the role of the Ca^2+^ signal in Purkinje cells during the movement in the lick task is needed. From the example mouse data in Figure 1B, it is difficult to tell how much of the behavioral response and neural activity is learned (a conditioned response) versus an unconditioned response to the water — it looks like the majority of the neural and behavioral response could be unconditioned. Which components of the lick task are cerebellum dependent-acquisition of learning? expression of the condition response? performance of unconditioned licks? Discussion of these issues would help to clarify the potential role of the calicum transients in the behavior.

We have added to the Discussion section to address some of these points. In summary though, we have not yet identified which aspects of lick-related behavioral refinement are due to this region of the cerebellum nor do we understand the importance of the climbing fiber-evoked responses in PCs, elicited at licking onset, to task performance. It is enticing to speculate on the potential role of these PC Ca^2+^ signals in guiding learning. But we conjecture that learning is not apt to be occurring during this well-practiced motor behavior. That said, climbing fiber activity has been shown to be important for motor memory retention (Medina et al., 2002). It may be that the climbing fiber-evoked Ca^2+^ signals that we studied are import for maintenance of some earlier aspect of behavioral adaptation during task training (e.g., timing of lick responses to the cue). Certainly, there is an abundant literature indicating that complex spikes evoked by climbing fibers are important for online motor control. As discussed, MLI-mediated suppression of nonlinear Ca^2+^ signaling could allow climbing fibers to multitask. Supporting both online motor control through the influence of complex spike bursts at the PC soma separate from the acquisition of plasticity in the dendrite.

d) In Figure 5 and Figure 6, might the lack of supralinear calcium responses result from the use of a suboptimal parallel fiber-climbing fiber pairing interval?

We tried different pairing intervals. However, supralinear Ca^2+^ responses were not apparent with feed-forward inhibition intact. These new data are now included in the manuscript (Figure—figure supplement 2).

[Editors' note: further revisions were requested prior to acceptance, as described below.]

[…] Essential revisions:

*1) The manuscript provides convergent evidence that inhibition from the molecular layer interneurons (MLIs) can gate supralinear calcium responses to combined parallel fiber and climbing fiber activity. This is likely correct. However, it is also difficult to rule out the possibility that* in vivo*, the chemogenetic suppression of MLIs could have resulted in changes (increases) in granule cell/parallel activity, which could contribute to the enhanced calcium responses. The authors need to be more forthcoming and explicit in the manuscript about acknowledging this possibility and the limited extent to which their evidence addresses it.*

In the revised manuscript, we now attempt to make these points more explicit and state the limitations of our approach/conclusions.

Their argument that argument that parallel fiber activity is not altered by suppression manipulation of MLI activity hinges on:a) new data showing that in the absence of MLI manipulation, calcium imaging-based measurement of parallel fiber activity closely tracks MLI activity (Figure 1—figure supplement 1) and an assertion in their point-by-point response that "unilateral chemogenetic disinhibition of MLIs in Crus II did not affect the population response of MLIs (data not shown). There are some concerns with this argument. First, just because the activity of two populations of neurons is similar under one measurement condition, it does not mean that it will be similar under all conditions, such as experimental manipulation of inhibition. Second, the whole point of the chemogenetic manipulation of MLIs is to suppress their activity, so why does it not affect the population response? Are we missing something? Third, the Kit promoter has some expression in Golgi cells, which directly inhibit the granule cells-this needs to be acknowledged.

We now state/discuss these caveats. The chemogenetic inhibitor hM4d is thought to work by a presynaptic action (now cited in the discussion, Stachniak et al., 2014). This would block release of GABA from MLIs, but not necessarily affect the calcium activity driven in large part by parallel fiber input. The revision includes specific mention of possible Golgi cell expression of hM4d in our *Kit::Cre* driver line.

b) Lack of change in the behavior. The additional data provided in Figure 2—figure supplement 3 is helpful, but does not fully address the possibility of other, unmeasured behavior differences (lateral deviation of the tongue, other orofacial movements), which is why the reviewers had asked for sample videos with and without chemogenetic suppression. Also, the lack of behavioral difference does not rule out a difference in the granule cells that does not affect the behavior, but could contribute to calcium transients in the Purkinje cells.In the absence of more direct recordings from the granule cells or parallel fibers comparing responses in the presence or absence of the disinhibition manipulation, we would be satisfied with a more thorough acknowledgement and discussion of the above caveats in the manuscript.

We now point this out in the Results section. Specifically we state that we cannot rule out behavioral alterations such as those described by the reviewer.

2) Figure 1—figure supplement 1B and 1E show that PF activity is well-correlated with licking behavior after the cue. However, it is not very clear in Figure 1—figure supplement 1D. In particular, fluorescence signals begin to increase mostly before licking starts in the B3 bouton. It is therefore important to show PF and MLI activity not only when the cue induces licking but also fails to induce licking as described above. In addition, please indicate in Figure 1—figure supplement 1D when the cue was presented. Otherwise, readers cannot tell which licking bouts are learned behavior.

The data for PF activity have been added to Figure 1—figure supplement 1. It is essentially a flat line. The result for MLIs is the same, but has been excluded from the plot for clarity. This is stated in the figure legend. The timing of the sound cue is now shown in panel D.

3) In the first round of review, one of the reviewers asked more discussion of the cued licking task, for example, which components of the lick task are cerebellum dependent-acquisition of learning. The authors address the comment, but their rebuttal letter has addressed the comment better than the manuscript itself. It is important to let readers know that it is currently unclear (1) the role of climbing fibers in the cued licking task, and (2) which aspects of this learning task are regulated by the cerebellum. These limitations do not diminish the value of this study.

Provided that the reviewers found our response in the rebuttal informative, we simply moved these sentences into the Discussion section of the revision.